

1    **Investigation of short-term effective radiative forcing of fire aerosols over North America**

2                                **using nudged hindcast ensembles**

3    Yawen Liu[1,2], Kai Zhang[2], Yun Qian[2], Yuhang Wang[3], Yufei Zou[3], Yongjia Song[3], Hui Wan[2],

4                                Xiaohong Liu[4], and Xiu-Qun Yang[1]

6                    [1] School of Atmospheric Sciences, Nanjing University, Nanjing, China

7                    [2] Pacific Northwest National Laboratory, Richland, Washington, USA

8    [3] School of Earth and Atmospheric Sciences, Georgia Institute of Technology, Atlanta, Georgia,

9                                USA

10   [4] Department of Atmospheric Science, University of Wyoming, Laramie, Wyoming, USA

13                    Corresponding to: Yun Qian [Yun.Qian@pnnl.gov]



**Abstract**

Aerosols from fire emissions can potentially have large impact on clouds and radiation. However, fire aerosol sources are often intermittent and their effect on weather and climate is difficult to quantify. Here we investigated the short-term effective radiative forcing of fire aerosols using the global aerosol-climate model Community Atmosphere Model Version 5 (CAM5). Different from previous studies, we used nudged hindcast ensembles to quantify the forcing uncertainty due to the chaotic response to small perturbations in the atmosphere state. Daily mean emissions from three fire inventories were used to consider the uncertainty in emission strength and injection heights. The simulated aerosol optical depth (AOD) and mass concentrations were evaluated against in-situ measurements and re-analysis data. Overall, the results show the model has reasonably good predicting skills. Short (10-day) nudged ensemble simulations were then performed with and without fire emissions to estimate the effective radiative forcing. Results show fire aerosols have large effects on both liquid and ice clouds over the two selected regions in April 2009. For the 10-day average, we found a large ensemble spread of regional mean shortwave cloud radiative effect over Southern Mexico (15.6%) and the Central U.S. (64.3%), despite that the regional mean AOD time series are almost indistinguishable during the 10-day period. Moreover, the ensemble spread is much larger when using daily averages instead of 10-day averages. For the case investigated here, a minimum of 9 ensemble members is necessary to get a reasonable estimate of the ensemble mean and spread of the forcing on individual days. This demonstrates the importance of using a large ensemble of simulations to estimate the short-term effective aerosol radiative forcing.



## 1. Introduction

Natural and human-induced fires play an important role in the Earth system. Aerosol and gas emissions from biomass burning can change the atmospheric composition and potentially affect the weather and climate. Over 30% of the global total emission of black carbon (BC) comes from open burning of forests, grasslands and agricultural residues (Bond et al. 2013). For organic aerosols, substantial increases of concentrations dominated by organic carbon enhancements are observed in regions with biomass burning events (Zeng et al. 2011; Lin et al. 2013; Brito et al. 2014; Reddington et al. 2014). As a result, biomass burning emissions have a large impact on the global and regional mean aerosol optical depth (Jacobson, 2014).

Through interactions with radiation and clouds, fire aerosols can significantly affect the long-term Earth's energy budget. Previous studies have investigated the global and regional radiative forcing of fire aerosols using long climatological simulations or satellite retrievals. For example, Ward et al. (2012) investigated the radiative forcing of global fires in pre-industrial, present day, and future periods. For the present-day condition, they estimated a direct aerosol effect (or radiative forcing through aerosol–radiation interactions as defined in IPCC AR5, RFari; see section 2.4) of $+0.1\,\mathrm{W\,m^{-2}}$ and an indirect effect (radiative forcing through aerosol–cloud interactions, RFaci) of $-1.0\mathrm{W\,m^{-2}}$. Using a newer model, Jiang et al. (2016) found similar RFari but slightly smaller RFaci ($-0.70\,\mathrm{W\,m^{-2}}$). Sena et al. (2013) assessed the direct impact of biomass burning aerosols over the Amazon basin using satellite data. Over the 10-year studied period, the estimated radiative forcing is about $-5.6\mathrm{W\,m^{-2}}$.

On short timescales, fire aerosols have even larger radiative impacts. Observed maximum daily direct aerosol radiative effects can reach $-20\mathrm{W\,m^{-2}}$ at TOA locally in Amazonia during biomass burning seasons (Sena et al., 2013). Very large direct effects of fire aerosols were



observed during extreme fire events over Central Russia (Tarasova et al. 2004; Chubarova et al.
2008; Chubarova et al. 2012). Instantaneous radiative effects of emitted aerosols reached -167
$W\,m^{-2}$ and monthly mean radiative effects reached about -65 $W\,m^{-2}$ in the 2010 Russia
wildfires (Chubarova et al. 2012). Using satellite data and a radiative transfer model, Kaufman et
al. (2005) found a radiative effect of -9.5$W\,m^{-2}$ due to smoke aerosol-induced cloud changes over
Southeast Atlantic for the 3 months studied.   Smoke-derived cloud albedo effect on local
shortwave radiative forcing is estimated to be between -2 and -4 $W\,m^{-2}$ in a day case study of
aircraft-measured indirect cloud effects (Zamora et al., 2016). Kolusu et al. (2015) investigated
direct radiative effect of biomass burning aerosols over tropical Southern America. By
quantifying results from the first and second day of 2-day single-member forecasts in September
2012, they found the modeled biomass burning aerosols reduced all-sky net radiation by 8
$W\,m^{-2}$ at TOA and 15 $W\,m^{-2}$ at surface.

Previous modeling studies on the short-term fire aerosol effects mainly focused on aerosol

direct effects (e.g., Keil and Haywood, 2003; Chen et al., 2014; Kolusu et al., 2015), and only a
couple of studies investigated the indirect effects of fire aerosols (Lu and Sokolik, 2013). In
addition, to estimate the aerosol indirect effect, long simulations (multi-years, >5 years
preferred) are often needed to remove the noise, because aerosol life cycle and cloud properties
are affected by strong natural variability on different timescales (Bony et al. 2006; Kooperman et
al. 2012). To solve the problem, alternative methods have been proposed to help extract signals
with shorter simulations. For example, nudging (also called Newton relaxation method) can help
reduce uncertainties associated with natural variability by constraining certain meteorological
fields towards prescribed conditions. A robust estimate of global anthropogenic aerosol indirect
effects can be obtained on substantially shorter timescales (1-2 years) by implementing nudging



to constrain simulations with pre-industrial and present-day aerosol emissions toward identical
circulation and meteorology (Kooperman et al. 2012). When nudged towards re-analysis data,
Zhang et al. (2014) found constraining only the horizontal winds is a preferred strategy to
estimate the aerosol indirect effect since it provides well-constrained meteorology without
strongly perturbing the model's mean climate state. Another example is the use of representative
ensembles of short simulations to replace a typical long integration. Wan et al. (2014) explored
the feasibility of this method and showed that 3-day ensembles of 20 to 50 members are able to
reveal the main signals revealed by traditional 5-year simulations.
In this study, we performed month-long and 10-day nudged CAM5 simulations to investigate
the effects of fire aerosols on radiation and cloud processes on short time scales (less than two
weeks). Horizontal winds were nudged towards reanalysis to constrain the large-scale circulation
and to allow for more accurate model evaluations against observations. We also used daily mean
emissions from three fire inventories to consider the uncertainty in emission strength and
injection heights. Even for short simulations, small perturbations of meteorological states might
have large impact on the local aerosol and cloud properties, thus bring uncertainty to the aerosol
forcing estimate. Therefore, in our simulations, we also employed very weak temperature
nudging in combination with ensembles to quantify the uncertainty.
The rest of the paper is organized as follows. Sect. 2 describes the model and data used in this
study. It also introduces how the ensembles are generated in the short nudged simulations and
explains how the fire aerosol forcing is estimated. Results and discussions are presented in Sect.
3 and conclusions are summarized in Sect. 4.
**2. Model, Method and Data**



## 2.1 Model description

In this study, we used the Community Atmosphere Model (CAM) version 5.3 with the finite volume dynamical core at 1.9° (latitude) × 2.5° (longitude) horizontal resolution with 30 vertical layers. The aerosol life cycle is represented by using the modal aerosol module MAM3 (Liu et al., 2012). CAM5 links the simulated aerosol fields with cloud and radiation through interactions of the aerosol module with the cloud microphysics and radiative transfer parameterizations. The two-moment bulk cloud microphysics scheme from Morrison and Gettelman (2008) is used to track mass mixing ratios and number concentrations of cloud droplets and ice crystals in stratiform clouds. Representation of shallow convection is based on the work of Park and Bretherton (2009). The deep convection parameterization was developed by Zhang and McFarlane (1995) and later revised by Richter and Rasch (2008) and Neale et al. (2008). Longwave and shortwave radiative transfer are calculated with the Rapid Radiative Transfer Model for GCMs (RRTMG, Malwer et al. 1997; Iacono et al. 2008).

## 2.2 Fire Emission Inventories

Three fire emission inventories were used in this study. Two of them are widely used bottom-up inventories— Global Fire Emissions Database version 3.1 (GFED v3.1, van der Werf et al., 2010; https://daac.ornl.gov/cgi-bin/dsviewer.pl?ds_id=1191) and GFED v4.1s (Giglio et al. 2013; Randerson et al. 2012; https://daac.ornl.gov/VEGETATION/guides/fire_emissions_v4.html). Another one is a top-down emission inventory—Quick Fire Emissions Dataset version 2.4 (QFED v2.4). GFED v3.1 and GFED v4.1s provide global monthly emissions at 0.25×0.25 degree spatial resolution from 1997 through the present. Daily emission data are obtained by disaggregating monthly emissions





based on daily temporal variability in fire emissions derived from MODIS measurements of
active fires (Mu et al. 2011). The more recent version GFED v4.1s improves by including small
fires based on active fire detections outside the burned area maps (Randerson et al., 2012).
QFED v2.4 estimates global fire emissions using the Moderate Resolution Imaging
Spectroradiometer (MODIS) measurements of fire radiative power and generates daily products
at 0.1×0.1 degree resolution.
To drive CAM5 simulations, fire emission data were regridded to the model resolution and
distributed vertically. For the GFED v3.1 and QFED v2.4 emission data we adopted the same
injection heights (from surface to 6 km) as used in the standard CAM5 model. While for
GFEDv4.1s, in this study the injection heights were estimated using a fire plume model and
scaled to the 6-hourly interval.
The fire emission inventories were first analyzed to select appropriate time periods and
regions for our study before being used to drive model simulations. Fig.1 shows the multi-year
mean biomass burning emissions from GFED v4.1 over North America. The emission manifests
significant seasonality with large dry matter consumption during March to April and June to
September. The summer and autumn burning covers Pacific Northwest and part of Canada and is
mainly associated with forest fires, while the spring burning occurs in more densely populated
regions like Mexico and central and eastern United States with a large contribution of
agricultural fires in croplands (Korontzi et al., 2006; Magi et al., 2012). Similar features are also
captured in GFED v3.1 and QFED v2.4 with differences in the magnitude. We chose to analyze
the simulated fire aerosol effect in April, the peak month of spring burning, when there are
extreme fire activities over Mexico (10 N to 25N, 100W to 80W) and occasionally large fires in
the Central U.S. (35 N to 45N, 100W to 85W). For the U.S., extended fire period is rare, making



it necessary to perform short-term evaluation. Fire aerosols formed from these two regions are
often transported to the Eastern and Southeastern U.S., where they mix with aerosols from
anthropogenic sources and potentially have significant impact on clouds and radiation over these
areas. Time series of regional mean fire emissions in April during 2003-2014 shows that
relatively large fires occur in both regions in 2009 (Fig.S1). Values of fire emissions in 2009 are
larger than the multi-year April mean by a factor of 1.9 in the Central U.S. and 1.5 in Southern
Mexico. Thus, in the following model simulations, we focused on analyzing the aerosol
properties and radiative effects over the two selected regions (denoted by the red boxes in Fig.1)
in April 2009.
Fire emitted BC from different emission inventories in April 2009 is shown is Fig.2. Although
GFED v4.1s includes the contributions of small fires (Randerson et al., 2012), the emitted BC in
GFED v4.1 shows no substantial increase compared to GFED v3.1during the selected period.
Only an increase by 1.75 is seen over Southern Mexico. In the Central U.S., the BC emission is
even slightly weaker in GFED v4.1. QFED v2.4 shows a much larger BC emission than the
GFED inventories. Values of emitted BC in QFED v2.4 are larger than those in GFED v4.1s by a
factor of 9.7 in the Central U.S. and a factor of 2.7 in Southern Mexico.
**2.3 Simulations**
Two groups of simulations were conducted (Table1) using the same greenhouse gas
concentrations, sea surface conditions and anthropogenic emissions of aerosols and precursors.
Each group includes four simulations, performed either without fire emission or with daily fire
emissions from one of the three fire emission inventories introduced in section 2.2. The emitted
species include BC, OC, and $SO_2$. Horizontal winds were nudged to 6-hourly ERA-Interim
reanalysis (Dee et al., 2011) as described in Zhang et al. (2014) in both groups.



Simulations in Group A are month-long single-member nudged simulations. These
simulations were performed to provide longer time series for model evaluation and generate
initial condition files for simulations in Group B. They started from January 1, 2009 and were
integrated for four months with 3-month spin-up. Initial condition files were generated on April
1 at 00 UTC for simulations in group B.
Simulations in group B are 10-day ensemble simulations. Unlike the traditional way of
perturbing initial conditions, in this study we constructed the ensembles by implementing a very
weak temperature nudging and perturbing the nudging time scale.  This is because under the
influence of horizontal-wind nudging, ensemble differences generated by perturbing initial
conditions would fade away during the integration. In contrast, our method can consider the
influence of small temperature perturbations during the entire simulation period, as nudging is
applied at every time step. On the other hand, the large-scale circulation patterns simulated in the
different ensemble members are very similar (not shown), so the noises caused by the chaotic
system can be constrained and the effective fire aerosol forcing signal can be easily identified.
Each ensemble in group B includes 10 members. The only difference between the members is
the relaxation time scale of temperature, which varies from 10 to 11 days at an interval of 0.1
day. All simulations started on April 1, 2009 and were integrated for 10 days. For each
simulation (e.g. E_QF), the initial condition was generated by combining the meteorological
fields from initial condition outputs in the S_NF simulation with aerosol and precursor
concentrations from initial condition outputs in the single-member simulation forced by the
corresponding fire emission (S_QF).
**2.4 Calculation of fire aerosol RF**





The IPCC AR5 report provides a more useful characterization of aerosol forcing by allowing for
rapid tropospheric adjustments (Boucher et al., 2013) compared to the original definition of
aerosol forcing. It quantifies aerosol radiative effects in terms of Effective Radiative Forcing
from aerosol-radiation interactions (ERFari) and Effective Radiative Forcing from aerosol-cloud
interactions (ERFaci). ERFari refers to the combined effect of instantaneous radiative forcing
from direct scattering and absorption of sunlight (aerosol direct effect) and related subsequent
rapid adjustments of atmospheric state variables and cloudiness (aerosol semi-direct effect).
ERFaci refers to the indirect forcing resulting from aerosol induced changes in cloud albedo
(cloud albedo effect) and subsequent changes in cloud lifetime as rapid adjustments (second
aerosol indirect effect) via microphysical interactions.
To allow for a straightforward comparison with previous studies in the literature, we followed
the IPCC concept of including rapid adjustments (effective aerosol radiative forcing), but
continued to decompose the aerosol effect in the conventional terms as aerosol direct radiative
effect (DRE), aerosol cloud radiative effect (CRE) and surface albedo effect. Note that as
nudging timescale determines the degree to which model physics are constrained (Kooperman et
al., 2012), the use of a 6-hour relaxation time scale for horizontal wind nudging means only very
fast adjustments are considered in the simulations.
Similar to Jiang et al. (2016), our calculations of fire aerosol DRE, CRE and surface albedo
effect are based on the work of Ghan et al. (2012) and Ghan (2013). They were calculated as the
radiative flux differences between simulations with and without fire emissions (denoted by $\Delta$). In
each simulation, aerosol (direct) forcing was defined as the difference between all-sky and clean-
sky TOA radiative fluxes ($F - F_{clean}$). Aerosol induced cloud forcing change was defined as the
difference between all-sky and clear sky TOA radiative fluxes under clean-sky



conditions $(F_{clean} - F_{clean,clear})$. The rest were related to surface albedo forcing $(F_{clean,clear})$.
Thus fire aerosol DRE, CRE, and surface albedo effect were expressed as $\Delta(F - F_{clean})$,
$\Delta(F_{clean} - F_{clean,clear})$, and $\Delta F_{clean,clear}$, respectively. More details about the method can be
found in section 2 of Ghan (2013). CRE includes contributions of both aerosol indirect effect and
aerosol semi-direct effect but was analyzed as a single term (i.e., the sum).
**2.5 Observational Data**
In this study, we used two sets of AOD reanalysis and the AERONET data (Holben et al.
1998) to evaluate the modeled AOD. The two AOD reanalysis datasets are the Naval Research
Laboratory (NRL) reanalysis (Rubin et al. 2015) and the Monitoring Atmospheric Composition
and Climate (MACC) reanalysis (Eskes et al. 2015). Both are generated by assimilating AOD
retrievals from MODIS (Zhang et al., 2008; Benedetti et al., 2009) with forecast fields. The NRL
reanalysis provides 6-hourly AOD at 1°horizontal resolution.  The MACC dataset provides 3-
hourly AOD at 1.125°horizontal resolution. Daily averages in April, 2009 were used for model
evaluation in this study. AERONET retrievals of AOD from April 1 to April 30 in 2009 were
used for model evaluation. Two sites are available in the selected regions: Cart Site (36°N,
97°W) and Mexico City (19°N, 99°W). LEV 2.0 cloud-screened all points AOD at 500 nm and
675 nm was used to generate hourly AOD at 550nm.
In addition, the simulated BC and primary organic matter (POM) concentrations were
compared with observations from the Interagency Monitoring of Protected Visual Environments
(IMPROVE) (Malm et al. 2004). IMPROVE aerosol data are only available over the Central U.S.
A total of fifteen sites were selected and marked in Fig 2, which include the sites west of 94°W
near the source region (asterisks) and sites east of 94°W in the downwind region (dots).





Observed organic carbon concentrations were multiplied by 1.4 for comparison with simulated

POM. Detailed descriptions about the data and sites are available at

http://vista.cira.colostate.edu/improve/. The IMPROVE network collect 24-hour aerosol data on

every third day. Daily averages during April, 2009 are compared on IMPROVE observation days

only.

**3. Results**

In this part, the model performance is first evaluated based on the simulations in group A.

Next, we present the simulated short-term effective fire aerosol forcing on 10-day and daily

timescales based on the results from group B simulations. We will demonstrate the importance of

using ensemble simulations in estimating the short-term aerosol effective forcing and give a

quantitative estimate of how many ensemble members are needed for the case selected in this

study.

**3.1 Model Evaluation**

Model simulated AOD are evaluated against the NRL and MACC reanalysis data (Fig. 3).

The simulated temporal variation of regional mean AOD over the central U.S. is consistent with

that in the reanalysis, but the magnitudes of simulated AOD are lower (Fig. 3). A better

agreement is found between the model and the NRL data, despite the horizontal winds in the

simulation are nudged towards a reanalysis that is very similar to the data used to derive MACC.

Temporal correlation coefficients (TCC) between the modeled AOD and the NRL reanalysis are

0.87 and 0.82 for S_QF and S_GF4 simulations, respectively, but are lower (0.67 and 0.78)

between the modeled AOD and the MACC reanalysis. The corresponding root mean square



errors rise from 0.13 (S_QF) and 0.1 (S_GF4) to 0.23 and 0.21. Generally, AOD is
underestimated by a factor of 2-4 in all simulations compared to the reanalysis, especially in
simulations with GFED emissions. Previous studies have suggested the need to scale up GFED
emissions by a factor of 1-3 to match the observed AOD (Tosca et al., 2013). This is consistent
with the large negative bias in the simulation S_GF3 and S_GF4. Simulated AOD in these two
simulations are almost indistinguishable due to the small difference in the total fire emission in
the region.
Over Mexico, different simulations produce similar temporal variations in AOD, but the
magnitude is smaller in the GFED simulations. Large discrepancies are found between model
results and reanalysis data during Apr. 17-20. An increase of AOD is captured by both reanalysis
datasets, while model results display a decrease of AOD compared to earlier days in the
simulation period. Note that the two sets of reanalysis data also have some differences
occasionally. For example, during Apr. 10-12, NRL data displays an increase of AOD, while
MACC data show the opposite. These discrepancies may partly result from the large internal
variability in this tropical region, where the simulated atmosphere state and its influence on
aerosol transport are more likely to disagree between the model and the reanalysis. Generally
speaking, the model forced with different fire emissions is capable of capturing daily variation of
AOD in both regions, especially during Apr. 1-10. This period was selected for further
investigation of the short-term fire aerosol effect.
Model simulated AOD are also evaluated against AERONET retrievals (Fig. 4). At Cart Site
(36°N, 97°W), with the QFED emission (S_QF) the model performs well in simulating both the
temporal variation (TCC=0.62) and magnitude of AOD. Simulations with GFED emissions also
reproduce the temporal evolution well (TCC = 0.58 for S_GF3 and 0.55 for S_GF4), but with



significantly low bias (mean bias by a factor of 2). The simulated difference in AOD magnitude
is similar to that found by Zhang et al. (2014) over the northern sub-Saharan African. Using the
QFEDv2.4 fire emission, the simulated regional mean AOD is a factor of 1.5 higher than that
using the GFEDv3.1 emission in their study.  Relatively good performance of S_QF is also seen
over Mexico. The simulated time evolution agrees well with AERONET retrievals except for
small discrepancies (e.g. during Apr.17 -19). A better agreement with the AERONET retrievals
is found for the NRL data than MACC reanalysis at both sites. Consistent with the evaluation
using reanalysis, the simulated temporal evolution of AOD during Apr. 1-10 agrees well with
both reanalysis data and AERONET retrievals in selected regions. This gives us further
confidence in choosing this period for further investigation.

The model is further evaluated against the IMPROVE data for BC and POM mass

concentrations (Fig. 5). In the downwind region, the simulated mass concentrations in simulation
S_QF lie within a factor of 2 of the observed values at most sites. However, the magnitude is
generally underestimated in simulations with the GFED emissions (S_GF3 and S_GF4),
especially in S_GF3. BC and POM concentrations in the downwind regions are affected by
transport of aerosols from Southern Mexico (Fig. S3). A larger amount of fire emission in
Southern Mexico would result in a higher BC (POM) concentration in the downwind region.
This explains the slightly higher concentrations in the simulation S_GF4 than S_GF3, as BC and
POM emissions over Southern Mexico are higher in GFED v4.1 due to the inclusion of small
fires (Randerson et al., 2012). The good agreement between S_QF and observations suggests that
the QFED data have a reasonable total emission rate. However, in the source region, the
simulation S_QF displays large positive bias with a large majority of the values fall out of the a-
factor-of-2 band. Given the reasonable total emission rate in QFED and a good agreement of



AOD with AERONET retrievals at Cart Site, this might result from the discrepancies in the
vertical distribution of the fire emissions. Fire-emitted BC and POM in simulations S_QF and
S_GF3 reach maximum values in the lowest level and decrease sharply to the next level, while
low-level fire emissions in S_GF4 distribute in a more uniform way (Fig. S4). As the sampling
was done on the lowest model level at most sites to compare with the IMPROVE data, this
explains the strong overestimation in S_QF. Although the same impact from vertical distribution
of fire emission also appears in simulation S_GF3, it is partly offset by its negative bias in the
total emission rate.

### 320    3.2 10-day Mean Results

Given the good model performance during April 1-10, we proceed to analyze the short-term
effects of fire aerosols during this period with nudged ensemble simulations. We define "fire
AOD" as the AOD difference between the simulations with and without fire emissions.

### 324    3.2.1 Fire Aerosol Distribution

Fig. 6 shows the spatial distributions of 10-day average ensemble mean fire AOD. For
reference, the total AOD in the simulation without fire emissions is shown in Fig. S2. During the
period, regional mean AOD increases by 6.4% (E_GF3), 6.4% (E_GF4) and 70.2% (E_QF) in
the central U.S. and 10.4% (E_GF3), 13.3% (E_GF4), and 49.6% (E_QF) in Southern Mexico
when fire emissions are included. In E_QF, high fire AOD covers almost the entire selected
region and extends further north. Maximum values of fire AOD stay above 0.2 around the
Yucatan Peninsula. Over the Central U.S, significant fire AOD ranging between 0.04 and 0.1
appears in the southwest part of the selected region. Apart from the significant AOD difference
in selected regions, large fire AOD also appears near the eastern coast as a result of local fire



emission and the eastward transport of fire aerosols from both regions. Overall, the modeled fire
AOD is much smaller in simulations with GFED emissions.
**3.2.2 Fire Aerosol Radiative Effect**

As described in Sect. 2.4, fire aerosol radiative effect can be decomposed into three items

including fire aerosol DRE, fire aerosol CRE and fire aerosol surface albedo effect.  Fig.7 shows
the spatial distributions of shortwave direct effect (SDRE) and shortwave cloud radiative effect
(SCRE). They are major contributors to the total fire aerosol forcing in the selected regions. For
reference, total aerosol forcing and total shortwave cloud forcing in the simulation without fire
emissions are shown in Fig. S2. The spatial distribution of SDRE and SCRE are similar for the
three cases, but with different magnitudes and statistical significant regions for simulations with
QFED and GFED fire emissions. In the Central U.S., fire aerosol SDRE is negligible in GFED
forced simulations due to small fire AOD. Although the fire AOD is larger in simulation E_QF,
the compensation between warming effect of fire BC and cooling effect of fire POM still results
a weak forcing of about -0.1W m$^{-2}$. Over southern Mexico, all simulations produce significant
cooling by fire aerosol SCRE with maximum values three times as large as those of
corresponding SDRE. For both SDRE and SCRE, the largest fire aerosol effects appear in the
E_QF simulation while the E_GF3 yields the weakest forcing, which is consistent with the
modeled fire AOD in these simulations.

In the following analysis, we will focus on the results from the E_QF simulation. Both SDRE

and SCRE spread outside the two selected regions and extend eastward reaching coast regions. A
stronger fire aerosol effect is seen in the Southern Mexico region. Strong SDRE appears over the
Yucatan Peninsula where fire AOD peaks (Fig. 6). Regional mean 10-day average of SDRE and
SCRE reach -1.66W m$^{-2}$ and -3.02W m$^{-2}$ respectively. In the central U.S, despite moderate fire



aerosol SDRE, SCRE near fire source region is weaker than -4 W m$^{-2}$, which is comparable to
that in the extended regions.

Given the largely insignificant change in cloud fraction (Fig. 8), fire aerosol SCRE in both

regions are mainly induced by changes in liquid water path (LWP) and droplet number
concentrations (CDNC). Changes in ice water path (IWP) and ice crystal number concentration
(ICNC) can also significantly affect SCRE, but with an opposite sign and mostly in the central
U.S.  Fire aerosol SCRE in the central U.S is associated with significant increases in both
column-integrated droplet number concentration (smaller droplet effective radius) and LWP,
indicating important contributions of both the aerosol first and second indirect effects. Increased
CDNC enhances cloud albedo by decreasing droplet sizes (Twomey, 1977) and allows more
liquid water to accumulate by decreasing precipitation efficiency (Albrecht, 1989; Ghan et al.,
2012). Note that although LWP and CDNC over southern Mexico change in a smaller magnitude
than those in central U.S., fire aerosol SCRE is stronger over Southern Mexico. This is mainly
due to the reductions in IWP and ICNC over the Central U.S. These changes, which possibly
caused by fire aerosol-induced changes in the circulation (Ten Hoeve et al, 2012), lead to a
positive SCRE that partly offsets the negative SCRE caused by changes in warm clouds. In the
northeast of the extended coastal regions, a more significant change of LWP comparable to that
in the central U.S appears, while a more significant change of CDNC comparable to that in
Southern Mexico occurs in the southwest. The combined effect leads to the total fire aerosol
effect in the extended regions.

The ensemble method provides another effective way to distinguish fire aerosol radiative

effect by comparing the radiative forcing distribution of ensemble members between simulations
with and without fire emission. A significant difference in the distribution of total aerosol (cloud)



forcing indicates a significant fire aerosol direct (cloud) effect. As shown in Fig. 9, a shift
towards stronger magnitude occurs to the total aerosol forcing when fire aerosols are considered.
Simulation E_QF has a larger percentage of grid cells with SDRE below -4.2W m$^{-2}$, while more
grid cells exceed -4.2W m$^{-2}$ in E_NF, which indicates a significant negative fire aerosol direct
effect. Same shift also appears to the total cloud forcing with more grid cells having cloud
forcing below -30W m$^{-2}$ in the simulation E_QF. Regional mean total aerosol and cloud forcing
in southern Mexico become more negative (-0.86 and -3.0 W m$^{-2}$) with fire aerosols.
Fig. 10 illustrates ensemble behavior of 10-day average regional mean total aerosol and cloud
forcing from all simulations as well as resulted fire aerosol SDRE and SCRE. The GFED forced
simulations not only resemble in ensemble mean, but also have small difference in ensemble
member distribution. Although members in the E_QF simulation capture stronger aerosol
forcing, thus stronger fire aerosol SDRE than those in E_GF3 and E_GF4, the ensemble spread
(as indicated by the maximum and minimum values) in the three simulations is similar.
Moreover, the E_QF simulation yields a smaller spread of SCRF compared with the GFED
forced simulations despite a stronger ensemble mean SCRF. In each fire simulation, ensemble
mean fire aerosol SCRE has a much larger magnitude than SDRE. So is the corresponding
ensemble spread. Taking results from E_QF simulation as an example, ensemble spread of
SCRE reaches 0.47 W m$^{-2}$, accounting for 15.6% of the corresponding ensemble mean, while
ensemble spread of SDRE is 0.03W m$^{-2}$ accounting for 3.5% of the corresponding ensemble
mean.
**3.3 Daily RF**
The fire aerosol effect is also investigated for individual days. The spatial distributions of
SDRE and SCRE on April 7 are shown in Fig 11, when relatively high fire emissions appear in



both regions.  Negative fire aerosol SDRE appears in the central U.S. biomass-burning region
indicating the dominant role of POM scattering. Fire aerosol SDRE over Southern Mexico shows
a contrast of warming effect in land region and cooling effect in adjacent ocean despite similar
aerosol loading in the two regions. However, they do have nearly equal clear-sky BC absorption
and POM scattering (Fig. 12). Difference in the low-level cloud distributions between two
regions leads to different signs of the simulated all-sky SDRE. Over land, when clouds appear
under elevated aerosol layers, more solar radiation is reflected back to space and this leads to
amplified BC absorption and more positive direct aerosol forcing (Keil and Haywood, 2003;
Zhang et al., 2016; Jiang et al., 2016). In contrast, neither absorption nor scattering changes
significantly from clear-sky to all-sky condition over adjacent areas over the ocean, since the
cloud fraction is small. Same enhanced absorption of above-cloud aerosols is also found over the
west Atlantic Ocean. Fire aerosols produce remarkable negative SCRE up to -16W m$^{-2}$ over
Southern Mexico land in response to the increase in CDNC and LWP.

**3.4 Discussion about the Simulation Strategy**

Fig. 13 shows the daily variation of the regional mean total (direct) aerosol forcing and cloud

forcing. Both the ensemble mean and spread are investigated here. The total aerosol forcing
exhibits considerable diversity across ensemble members within each simulation, even though
the simulated AOD is nearly indistinguishable (Fig. 3). Taking results from simulation E_QF as
an example, maximum values of difference between members exceed 0.4 W m$^{-2}$ for aerosol
forcing and 5 W m$^{-2}$ for cloud forcing, which are approximate 10% of the corresponding
ensemble mean values.  The large spread of total aerosol forcing and cloud forcing will lead to
uncertainties in the estimation of fire aerosol effect. This points out the importance of conducting
ensemble simulations in order to get a more comprehensive estimate of the daily fire aerosol



effect. The minimum ensemble size required for this case is investigated in terms of the

ensemble mean and spread estimate. Simulated ensemble mean fire aerosol SDRE remains

nearly unchanged regardless of the ensemble size (Fig. 14a). However, discrepancies in the

ensemble mean fire aerosol SCRE (Fig. 14b) are substantial when the number of ensemble

members is smaller than 8. The same is true for the ensemble spread of fire aerosol SCRE (Fig.

S5). Overall, the time evolution and magnitude of ensemble mean and spread tend to converge

when the number of ensemble members reaches about 9 for different days we investigated here.

Fire aerosol sources are often intermittent and height-dependent and there is a need to

estimate the short-term effective aerosol forcing. Although nudging helps to constrain large-scale

features, the simulated cloud properties (e.g. cloud fraction and LWP) and their response to

aerosol changes can still be sensitive to small perturbations in the atmospheric state. Therefore,

for investigating the short-term aerosol effect, a single simulation might not be sufficient to tell

whether the aerosol effect is significant. The use of ensembles provides an effective way to

estimate the uncertainty. Previous investigations of short-term fire aerosol effect are mainly

based on single-member simulations (Wu et al., 2011; Sena et al., 2013; Kolusu et al., 2015).

While this might be less a problem for SDRE, one should be more careful when investigating the

aerosol indirect effect and conduct ensemble simulations to see whether the estimated fire

aerosol effects are robust.

**4. Summary**

In this study, we investigated the short-term effect of fire aerosols on cloud and radiation

using CAM5 simulations. Month-long single-member simulations and 10-day ensemble

simulations were conducted in April 2009. In order to help extract signals on short time scales,



we used nudging to constrain horizontal winds in all simulations. Our investigation focused on
Southern Mexico where there were constant intensive fire activities and the Central U.S. with
occasionally large fires. Apart from the local effect, fire emissions from the two regions are
shown to affect downwind coastal regions through transport.

Modeled AOD and mass concentrations (BC and POM) were evaluated against observations.

In general, all simulations with fire emissions reproduce the observed temporal variation of daily
mean AOD well, although the simulated magnitude is smaller. The model performance is better
when QFEDv2.4 is used, which has larger fire emissions. Modeled regional mean AOD values in
simulations using two versions of GFED fire emission data are barely distinguishable, despite the
inclusion of small fires and changed injection heights in GFEDv4.1 used in this study. Both of
them simulate about a factor of 1.5 smaller AOD than that in the simulation using the QFED fire
emissions. At sites in the downwind region, the modeled BC and POM mass concentrations in
the simulation with QFEDv2.4 emission (S_QF) agree well with the IMPROVE data. In contrast,
simulations with the other two fire emission datasets (S_GF3 and S_GF4) have a low bias. The
simulated AOD in the source region in S_QF also agrees well with the AERONET data (Cart
Site). If there is no large compensating error in the model, QFEDv2.4 seems more reasonable in
terms of the total (vertically-integrated) emission rate. On the other hand, S_QF strongly
overestimates BC and POM concentrations in the source region. Considering that the source-
region AOD and the downwind surface mass concentrations are well simulated, the
overestimation suggests that the actual emission peak might appear at higher levels compared to
the height-dependent injection rates applied in the S_QF simulation.

Based on the evaluation, we chose the first 10 days as the simulation period and focused on

the simulation with QFEDv2.4 fire emission in our ensemble nudged simulations. In our method,





the nudged ensembles are generated by adding a very weak temperature nudging along with
horizontal-wind nudging and perturbing the nudging time scale of temperature gently. In this
way, small temperature perturbations are added to the simulation at each time step, while the
large-scale circulation features are very similar between individual members. We first
investigated the 10-day mean effective fire aerosol forcing. Decomposition of total aerosol
radiative forcing shows that fire aerosol effects in the two selected regions are dominated by the
shortwave cloud radiative effect SCRE.  All fire simulations show similar spatial distribution of
SDRE and SCRE, but with different magnitudes and statistically significant regions. The
similarity in the spatial distribution is expected since the three emission datasets differ mainly in
the emission magnitude and not much in spatial distribution in the focused regions of this study.
Fire aerosol effects in simulations with GFED emissions (E_GF3 and E_GF4) are weaker than
that with QFEDv2.4 emission (E_QF) by a factor of 1.5 for SCRE and a factor of more than 4
for SDRE. Generally speaking, the difference in simulated AOD and fire aerosol indirect
radiative effects between simulations is smaller compared to the difference between fire
emissions, consistent with the findings in sub-Saharan African biomass-burning region (Zhang et
al. 2014).
Fire aerosols produce a negative direct effect of -0.1 $\mathrm{W\,m^{-2}}$ in the Central U.S. and -0.86
$\mathrm{W\,m^{-2}}$ in Southern Mexico in E_QF during the 10-day period. Within each region, negative fire
aerosol SDRE peaks where fire AOD reaches maximum. Unlike the limited area affected by
significant fire aerosol SDRE, fire aerosol SCRE from selected regions spreads eastward and
northward, affecting remote coastal regions. Maximum SCRE stays below -4 $\mathrm{W\,m^{-2}}$ in the
central U.S. and -10 $\mathrm{W\,m^{-2}}$ in Southern Mexico in response to significantly increased LWP and
CDNC. Decreases of IWP and ICNC also contribute to fire aerosol SCRE in the Central U.S. but



with an opposite sign. The offset effect of the positive forcing induced by changes in cloud ice
properties explains the smaller SCRE in the central U.S. despite the larger changes in cloud
droplet properties.
We also investigated fire aerosol effects on the daily time scale, where the variation in the
simulated fire aerosol effect can be large among the ensemble members. The large ensemble
spread of total aerosol and cloud forcing indicates large uncertainties in estimating daily fire
aerosol effects, despite similar AOD across ensemble members. Further investigations show that
the simulated ensemble mean and spread with less than 7 members differs considerably to those
with more members. A minimum of 9 members is necessary to achieve a steady estimate of the
magnitude and temporal variation of SCRE in this case. Our results suggest that for short-term
simulations of aerosol and cloud processes, even small perturbations might result in large
difference across members despite constrained large scale features. In order to obtain a robust
estimate of the effective fire aerosol forcing during a short period, it is important to conduct
ensemble simulations with sufficient ensemble members.

**Acknowledgments**
This study was supported by the U.S. Department of Energy (DOE)'s Office of Science as part
of the Regional and Global Climate Modeling Program (NSF-DOE-USDA EaSM2). The work
was also supported by the National Natural Science Foundation of China (NSFC) under Grants
No. 41621005 and 41330420, the National Key Basic Research Program (973 Program) of China
under Grant No. 2010CB428504, and the Jiangsu Collaborative Innovation Center of Climate.
The Pacific Northwest National Laboratory (PNNL) is operated for DOE by Battelle Memorial



Institute under contract DE-AC05-76RL01830. Computations were performed using resources of
the National Energy Research Scientific Computing Center (NERSC) at Lawrence Berkeley
National Laboratory and PNNL Institutional Computing. All model results are available from the
corresponding author upon request.





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





Table 1. List of CAM5 simulations.

| Name | Fire emission | Simulation period | Member | Nudging |
|---|---|---|---|---|
| Group A: Single member simulations | | | | |
| S_NF | No | | | |
| S_GF3 | GFED v3 | January 1- April 30, 2009 | 1 | Horizontal winds (6h) |
| S_GF4 | GFED v4.1 | | | |
| S_QF | QFED v2.4 | | | |
| Group B: Ensemble simulations | | | | |
| E_NF | No | | | Horizontal winds (6h) and temperature (~10d)* |
| E_GF3 | GFED v3 | April 1 - April 10, 2009 | 10 | |
| E_GF4 | GFED v4.1 | | | |
| E_QF | QFED v2.4 | | | |

* See section 2.3 for details about ensembles



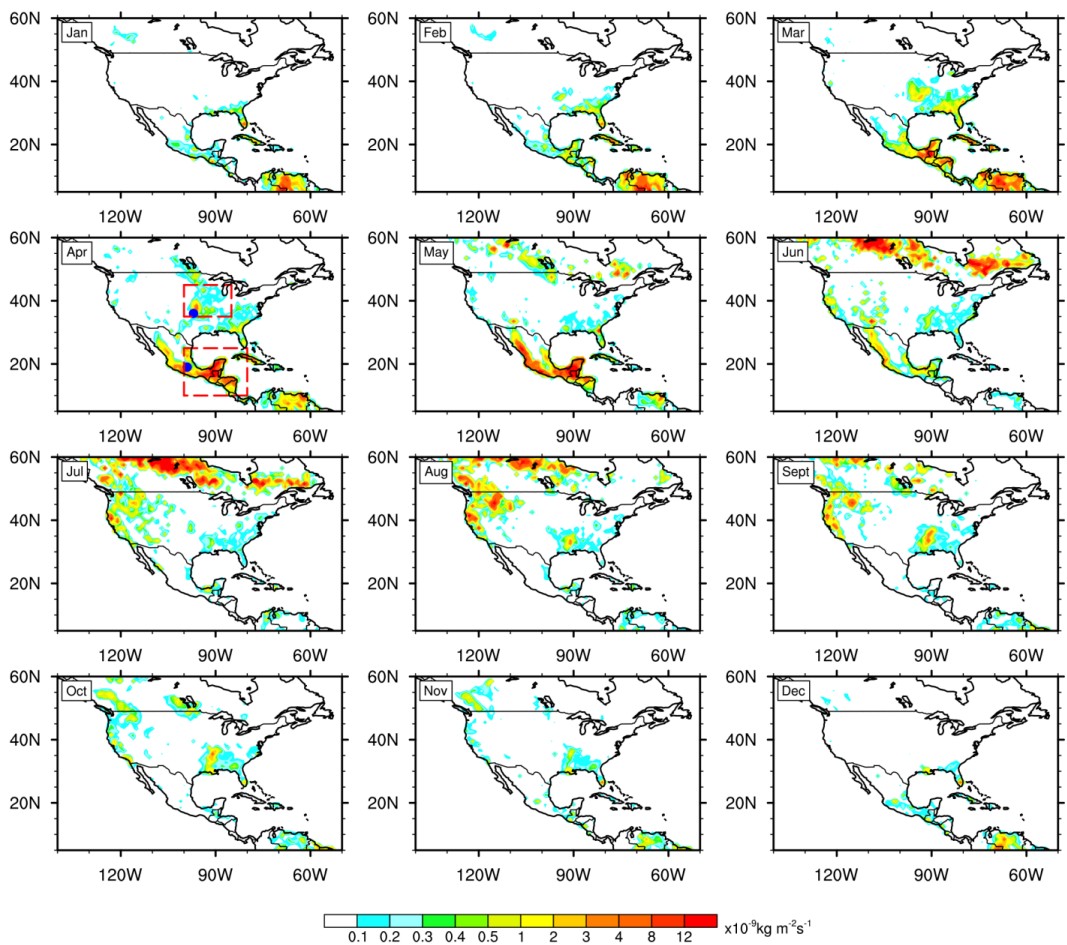

Figure 1. Spatial distributions of multi-year monthly mean biomass burning consumed dry matter over North America during 2003-2014 from GFEDv4.1. Boxes denote selected regions: central U.S (35 - 45°N, 85 - 100°W) and Southern Mexico (10 - 25°N, 80 - 100°W). Dots denote locations of AERONET sites: Cart Site (36°N, 97°W) and Mexico City (19°N, 99°W)




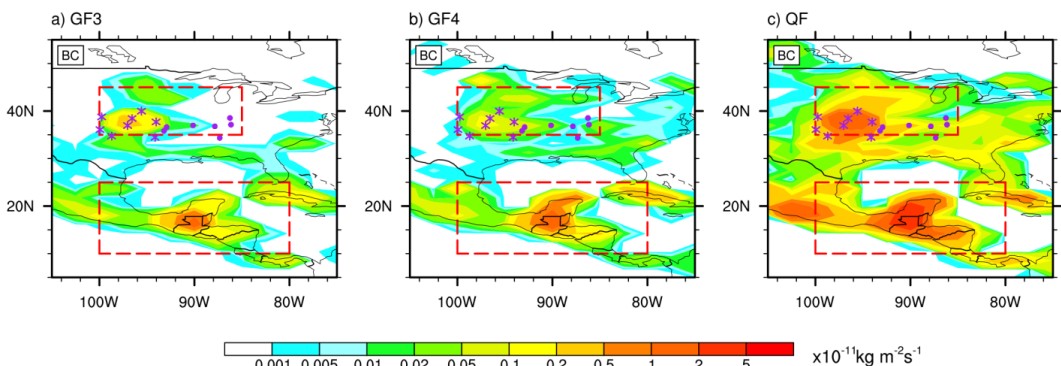

Figure 2. Spatial distributions of monthly mean BC emissions from three emission inventories in April, 2009. IMPROVE data sites are shown as asterisks for sites near the source region and as dots for sites in the region downwind of the fire source.





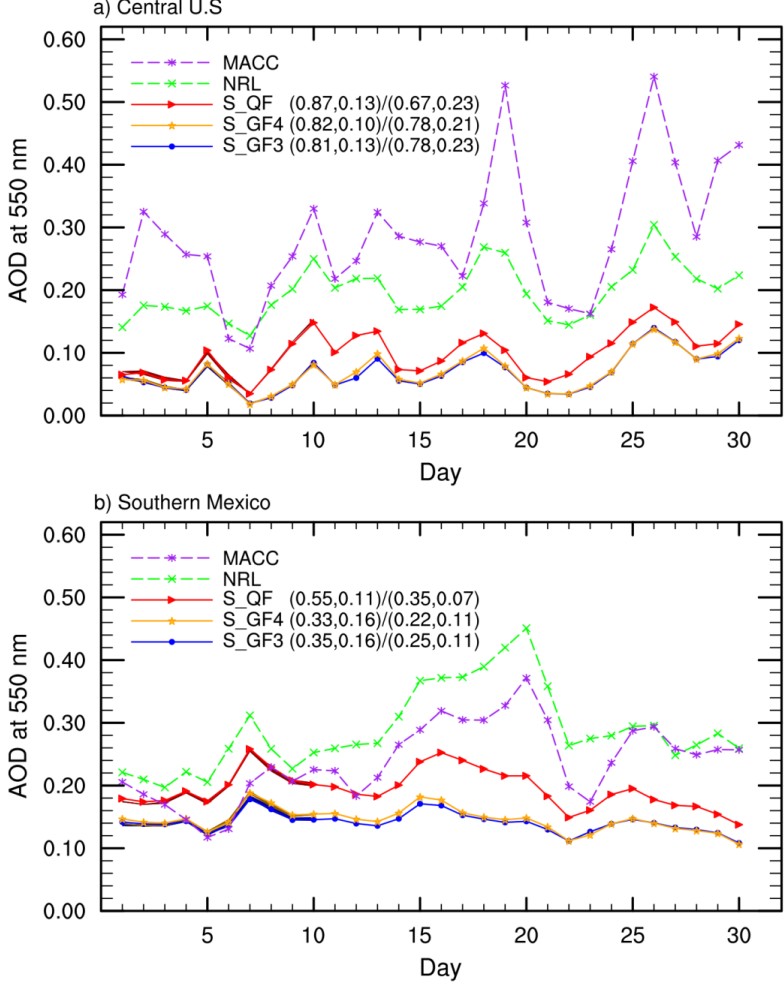

Figure 3. Time series of daily regional mean AOD in April, 2009 in simulations and reanalysis data. Numbers in parenthesis denote TCC and RMSE between each simulation in group A and reanalysis data (left: NRL; right: MACC). Individual lines indicate group A simulations. Shaded areas (very narrow) in slightly darker colors during April 1-10 illustrate maximum and minimum values of daily mean AOD among ensemble members in group B simulations.




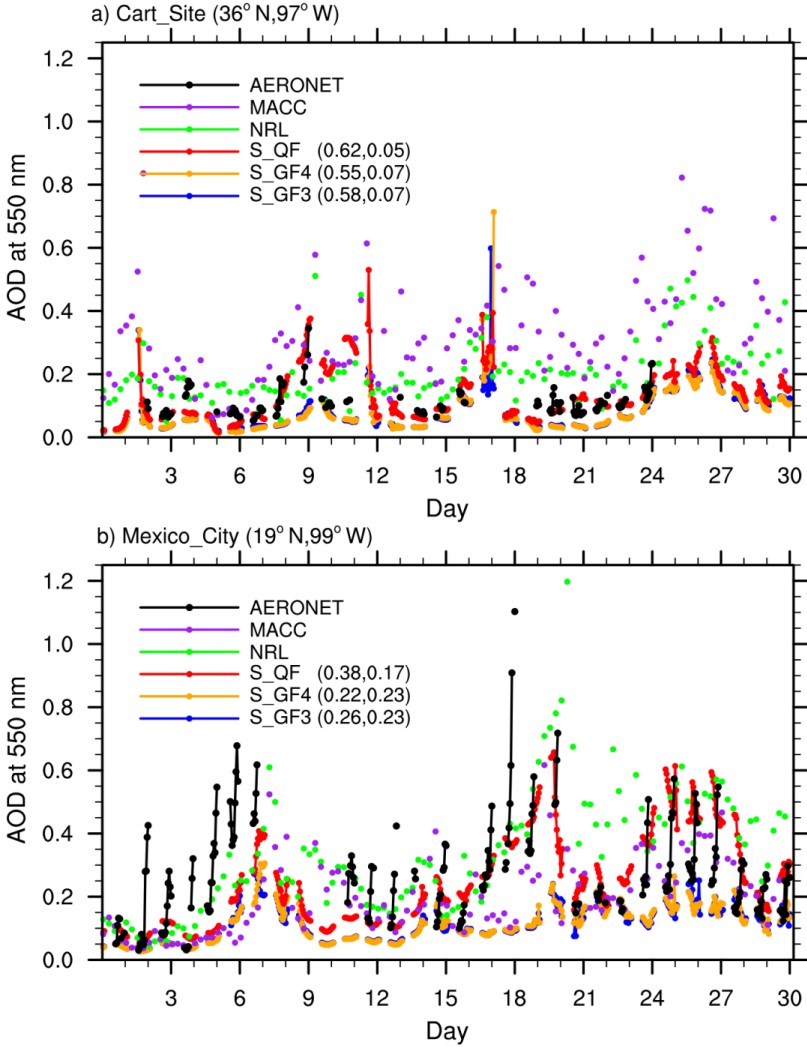

Figure 4. Time series of hourly regional mean AOD in April, 2009 from group A simulations, reanalysis data and AERONET retrievals at AERONET sites. Numbers in parenthesis denote TCC (left) and RMSE (right) between each simulation and AERONET AOD.




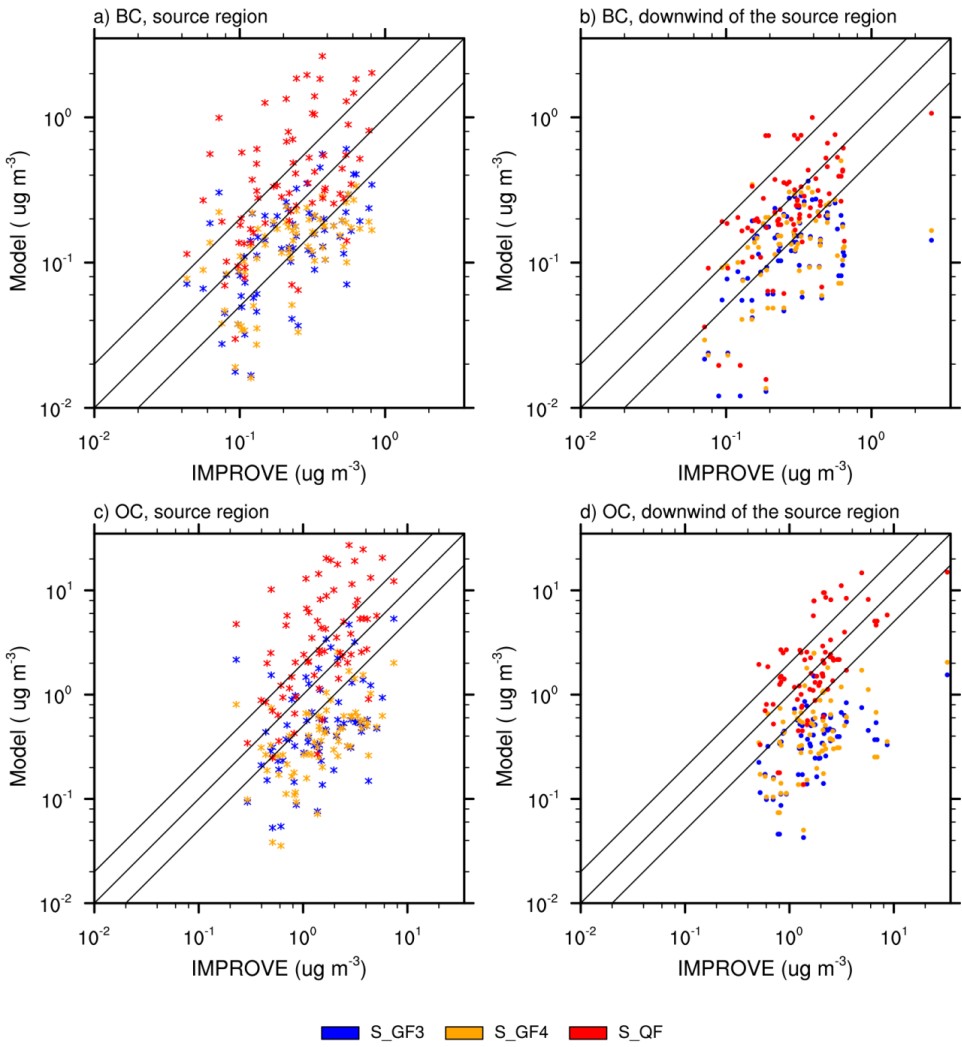

Figure 5. Evaluation of the simulated BC (up) and POM (bottom) concentrations in group A simulations against the IMPROVE data at sites near the source and downwind the source region. Locations of these sites are marked with the same symbol as in Fig. 2.





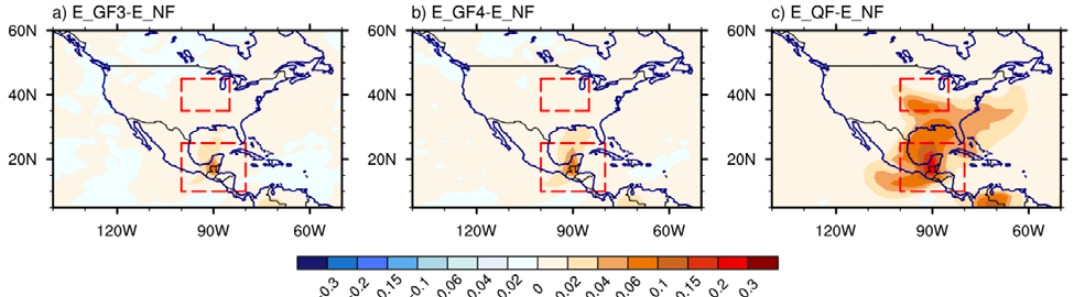

Figure 6. Spatial distributions of 10-day average (Apr. 1-10) ensemble mean AOD differences between simulations with (E_GF3, E_GF4, and E_QF) and without fire emission (E_NF).




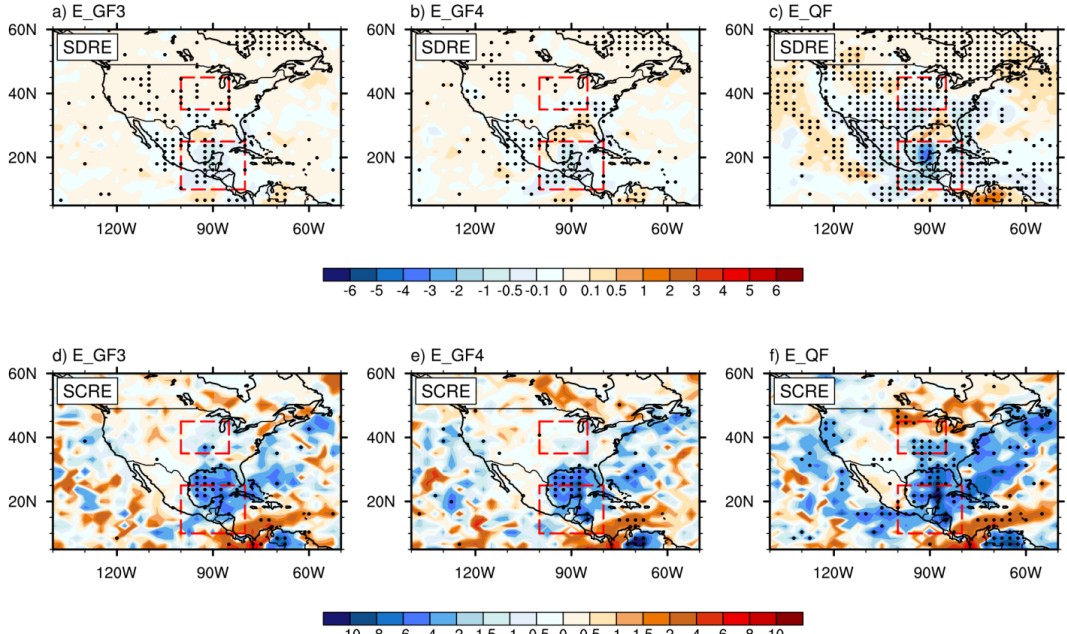

Figure 7. Spatial distributions of 10-day average (Apr. 1-10) ensemble mean fire aerosol shortwave direct radiative effect (SDRE) and shortwave cloud radiative effect (SCRE) ( W m$^{-2}$ ) in group B simulations. Dots denote regions where SDRE is statistically significant at the 95% confidence level based on the KS test.




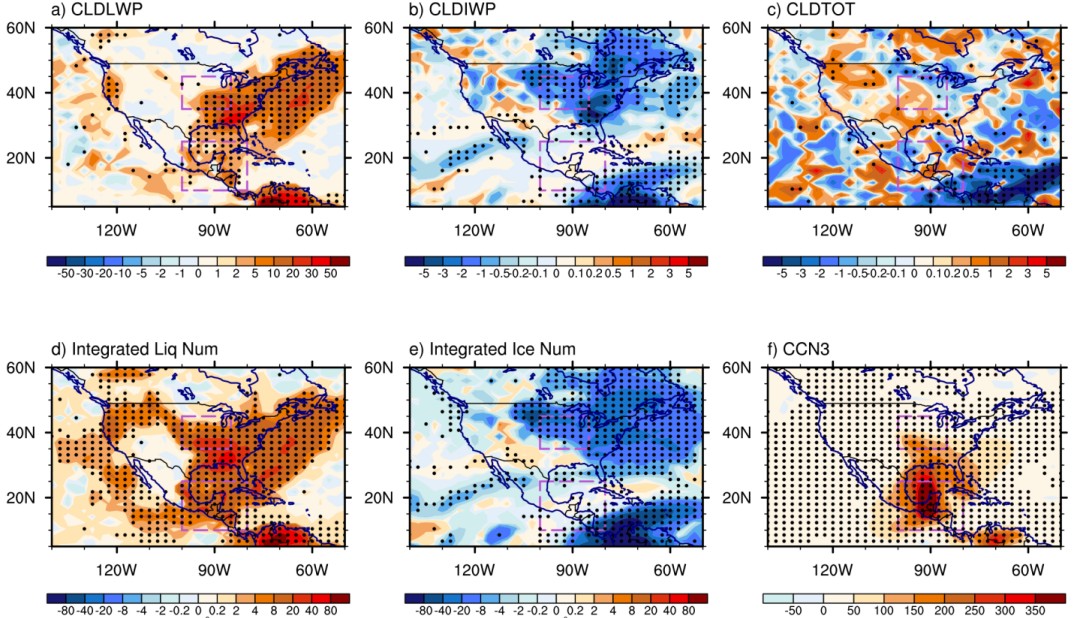

Figure 8. Difference of 10-day average (Apr.1-10) ensemble mean between simulations E_NF and E_QF: a) cloud liquid water path ( $g\,m^{-2}$ ), b) cloud ice water path ( $g\,m^{-2}$ ), c) total cloud fraction (%), d) column-integrated droplet number concentration ( $m^{-2}$ ), e) column-integrated ice number concentration ($m^{-2}$ ), and f) cloud condensation nuclei at 0.1% supersaturation near 900 hPa. Dots denote regions where the difference is statistically significant at the 95% confidence level based on the KS test.

.





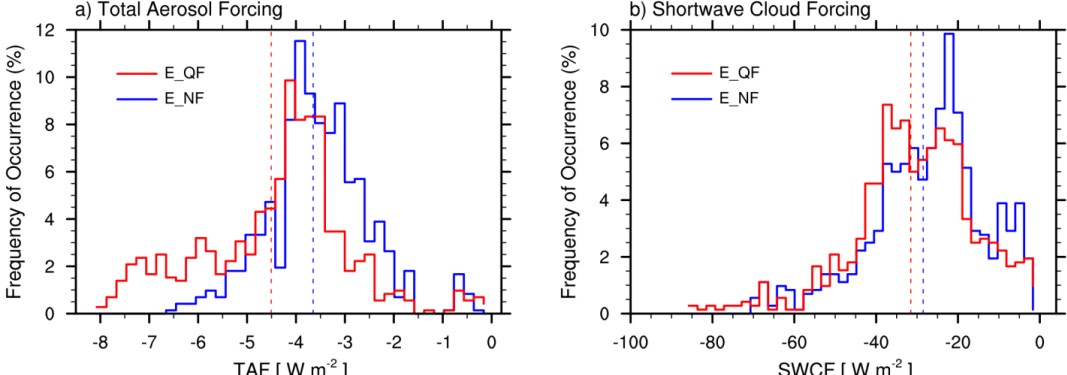

Figure 9. Probability distributions of 10-day average (Apr.1-10) a) total aerosol forcing and b) cloud forcing over Southern Mexico in simulations E_NF and E_QF sampled from grid values of ensemble members (72x10 for each case). Dashed lines indicate the mean of the distribution.



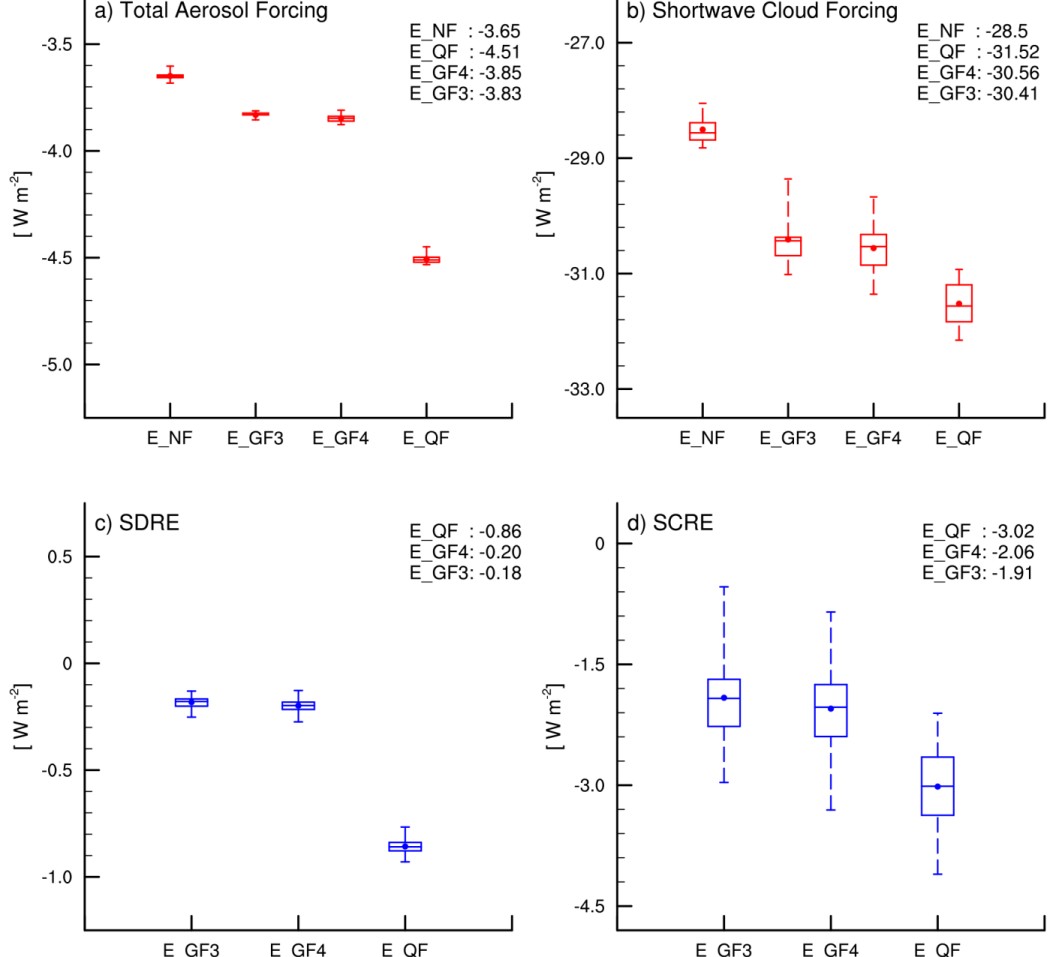

Figure 10. 10-day average (Apr. 1-10) regional mean a) total aerosol forcing, b) total shortwave cloud forcing and fire aerosol, c) SDRE, and d) SCRE in Southern Mexico in group B simulations. Box denotes the 25th and 75th percentiles. Bars outside the box indicate minimum and maximum. Bar within the box denotes the 50th percentile. Total aerosol and cloud forcing are sampled from different ensemble members (10 for each case). Fire aerosol SDRE and SCRF are sampled by calculating the difference between members in simulations E_QF (E_GF3/E_GF4) and E_NF (10x10 for each case).



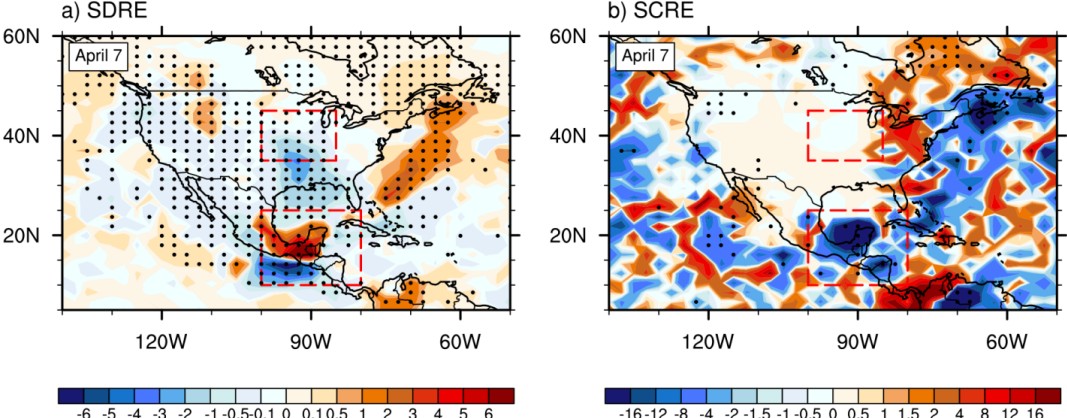

Figure 11. Spatial distributions of ensemble mean fire aerosol a) SDRE and b) SCRE ( W m$^{-2}$ ) on April 7 in the E_QF simulation. Dots denote grids where fire aerosol effect is statistically significant at the 95% confidence level based on the KS test.




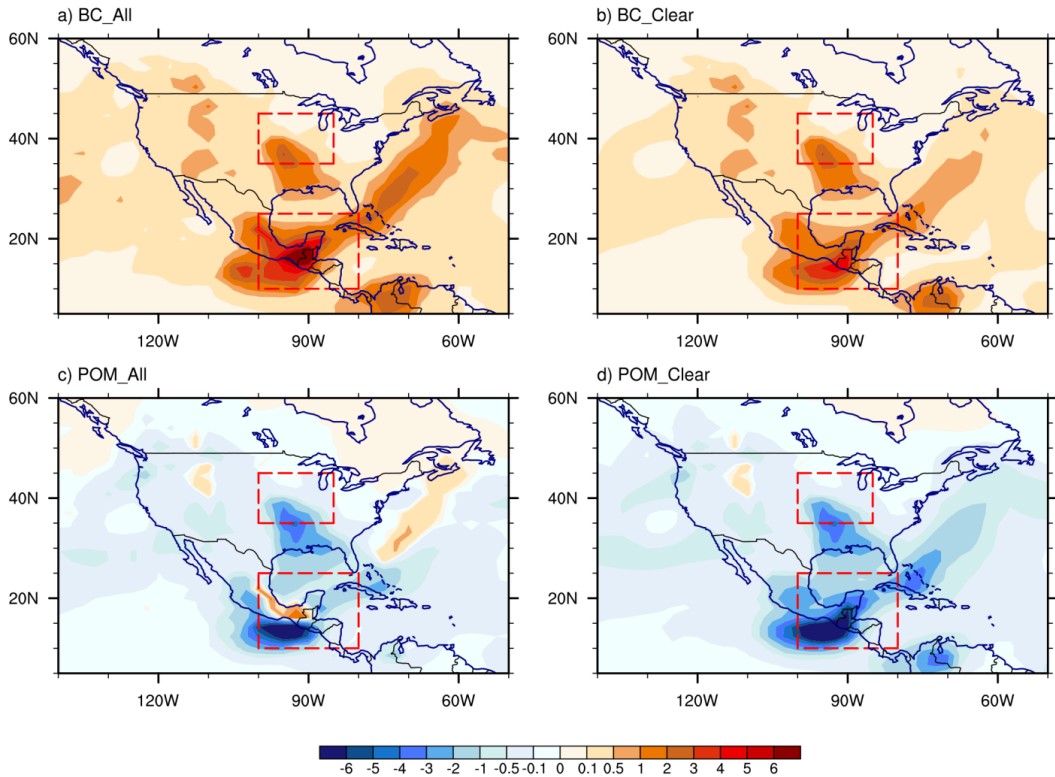

Figure 12. Spatial distributions of fire BC SDRE and fire POM SDRE ( W m$^{-2}$ ) on all-sky and clear-sky conditions on April 7 in the E_QF simulation.





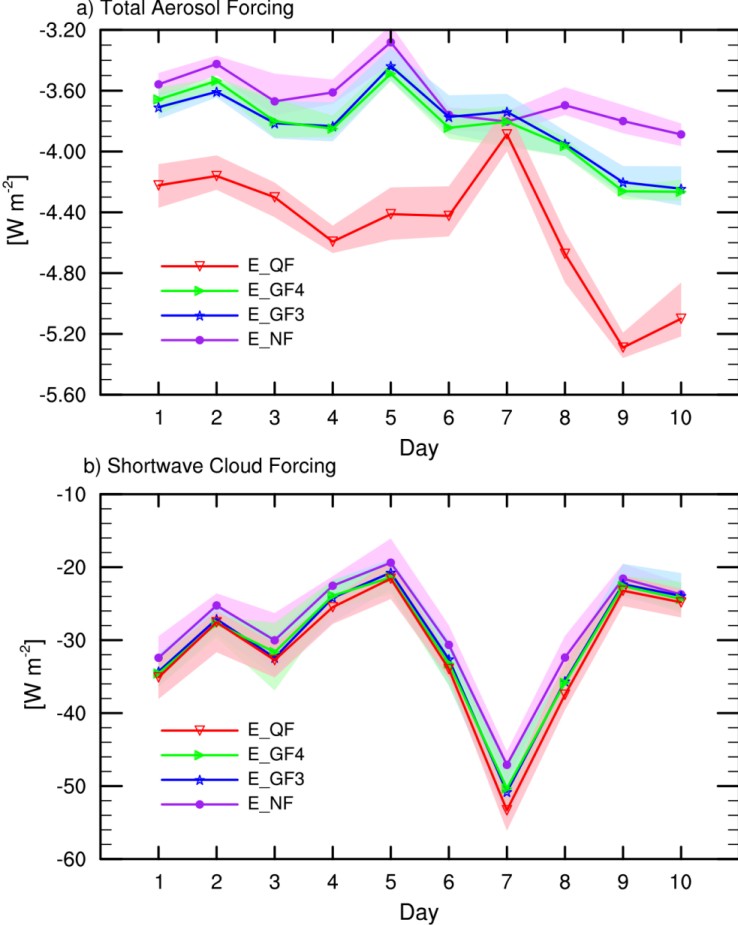

Figure 13. Time series of daily regional mean total a) aerosol forcing and b) cloud forcing in Southern Mexico during Apr.1-10, 2009 in group B simulations. Individual lines indicate ensemble mean values. Shaded areas illustrate the ensemble spread (from minimum to maximum).





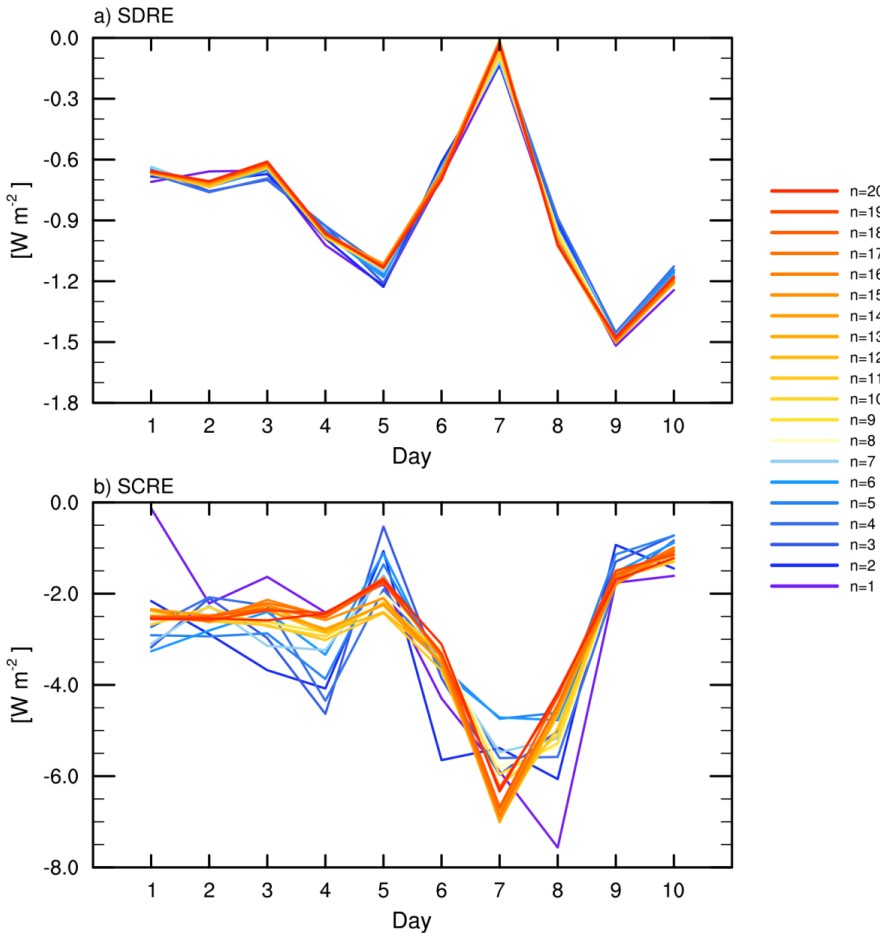

Figure 14. Time series of daily ensemble mean fire aerosol a) SDRE and b) SCRE averaged over
Southern Mexico during Apr. 1-10, 2009 in QFED forced ensemble simulations with varying the
total number of ensemble members (n=1-20).