# Peer review of "Investigation of short-term effective radiative forcing of fire aerosols over North America using nudged hindcast ensembles"

_Atmospheric Chemistry and Physics, 2017_

## Referee Comment (RC1) · Anonymous Referee #1 · 26 Jun 2017

The major problem is that you should explain your results, not just describe the figures. How is the CRE effect influenced by LWP and CDNC? Why does the non-local effect exist? Especially, why the CRE maximum occurs over the Mexican gulf. These should be discussed and investigated.

Minor comments: L60: studied->study L77-79: You should clarify why the indirect effect of fire aerosol deserves study. L101-103: What's the differences between nudging horizontal wind and temperature. L370: Add "are" between "which" and "possibly"; L384: Same->The same; L297-503: How you can get the conclusion that at least 9 members are needed from Fig. 14. You need to quantify the results.

---

## Referee Comment (RC2) · Anonymous Referee #2 · 8 Jul 2017

This papers studies the direct/indirect aerosol effect from fires using CAM5 with nudged horizontal wind speed and/or nudged temperature. Overall the approach is sound and the paper is well written. Yet it still needs some major clarifications before it is accepted for publication.

General comments:

Since fire emission inventories are critical to this study, please provide a table or a plot to show the BC/OC/SO2 emissions from the 3 different inventories quantitatively. The colorbar in Fig. 2 is difficult to tell how bigger is the QF than the GF3/4. It seems QF emissions are at least 5 times larger than the other two. Please provide a table showing the different fire aerosol forcing components. In the introduction part, 3 different aerosol forcing are mentioned, but only direct forcing and short wave cloud forcing are presented in the result section. Please show long wave forcing and surface albedo forcing as well in the table. The initial condition for the 10-day ensemble runs is generated by S_NF with only u and v being nudged. Temperature is not nudged for the S_NF run. So my question is when you now include slow temperature nudging in the ensemble runs, will they go through some adjustment through the 10-day period? Or in other word, how well is the initial T compared to the T being nudged to on April 1st?

Specific comments: Abstract: why no forcing numbers are provided here. It is expected to see direct forcing and net indirect forcing rather than some changes in the short wave cloud forcing.

Line 62-76: Please show what these forcings are. Direct or indirect?

Line 103 : please provide relaxation time for the very weak temperature nudging.

Line 131-132: please show or elaborate how you convert monthly mean emissions to daily emissios.

Line 138-142: Does the CAM5 default/background emission already include fire emission? Or did you remove fire emissions from the CAM5's emission files if there is any?

Line 220-226: please define these forcings with a few sentences rather than refer readers to Ghan 2013.

Line 238 : please explain what "LEV 2.0 cloud-screened" is.

Line 268-272: How much does the fire emitted aerosols contribute to the total AOD? It would be helpful to show some estimate of the contributions from fire emitted aerosols and other background aerosols. I realized you presented background AOD and fire AOD later in Fig. 6. But it would be more helpful if you can present some data here when you quote the need to increase the fire emissions by a factor of 1-3. And please explain why increasing the fire emission by a factor 1-3 could then make the simulated

[Figure]

AOD large enough to compare with the reanalysis. It seems it is still unlikely to me.

Line 357-358 : it is confusing here. Please consider revising.

Line 370-372 : In the simulations with nudged U and V, the circulation is constrained. So it seems the circulation change may be small enough. Then use this to explain to change of ice clouds. I suspect the coarse mode dust number may be smaller and this may contribute to the decrease of produced ice number since the ice nuclei number(dust) is smaller. Need further investigation here.

Line 385: make it clear it is SW cloud forcing.

Line 420: Why quote Fig. 3 here? I think Fig.3 shows results from Group A not from ensemble runs.

Line 430-432: How is the spread calculated for different N? Also how do you select the ensemble member for each different N? I suspect the number 9 required to converge may be different if the ensemble members for different N are constructed differently.

Fig 3 : Please give full name of TCC.

Fig 7 : what is is the KS test? Please give full name.

Fig 10: is a) total aerosol direct forcing?

---

## Author Comment (AC1) · 13 Sep 2017

Please find the detailed reply in the supplement zip file

Please also note the supplement to this comment:
https://www.atmos-chem-phys-discuss.net/acp-2017-414/acp-2017-414-AC1-supplement.zip

---

## Author Comment (AC2) · 13 Sep 2017

Please find the detailed reply in the supplement zip file

Please also note the supplement to this comment:
https://www.atmos-chem-phys-discuss.net/acp-2017-414/acp-2017-414-AC2-supplement.zip
* * *

---

## Author Response (AR1)

Reply to Anonymous Referee #1

We thank the reviewer for the careful review and helpful comments. Our responses are detailed below (reviewer's comments marked in blue and our responses in black.

**Comment:** The major problem is that you should explain your results, not just describe the figures. How is the CRE effect influenced by LWP and CDNC? Why does the non-local effect exist? Especially, why the CRE maximum occurs over the Mexican gulf. These should be discussed and investigated.

**Reply:** Following the suggestion, the corresponding paragraph has been reorganized and additional description is added. We now explain how changes in cloud droplet number concentration (CDNC) and liquid water path (LWP) result in the negative SCRE in detail. The non-local effect, that is, the tendency of maximum SCRE to appear over the Gulf of Mexico is related to a more sensitive SCRE response to the larger relative change of CDNC and LWP over Gulf of Mexico compared to the land region. As shown in Fig.8 in the original manuscript, changes in both CDNC and LWP are of comparable magnitudes between Gulf of Mexico and the land region. However, given the smaller background CDNC and LWP over Gulf of Mexico, SCRE is more sensitive to changes in the two items over Gulf of Mexico than in the land region. In the revised paper, we have pointed out this phenomenon (Line 315-316) and provided an explanation (Line 337-341).

It now reads (Line 315-316):

*"It's interesting to note that the maximum SCRE tends to center around adjacent Gulf of Mexico rather than the land region."*

and (Line 320-348 ):

*"To find out the causes of the fire aerosol SCRE, fire aerosol-induced changes in cloud properties are analyzed. Given the largely insignificant change in cloud fraction (Fig. 8), the negative fire aerosol SCRE in the selected regions is mainly associated with increases in cloud droplet number concentrations (CDNC) and liquid water path (LWP). The increased CDNC due to an increase of CCN from fire aerosols (Fig. 8) leads to smaller droplet sizes, which in turn increase cloud albedo*

*by enhancing backscattering (Twomey, 1977) and further affect LWP by decreasing precipitation efficiency and allowing more liquid water to accumulate (Albrecht, 1989; Ghan et al., 2012). These changes in warm cloud properties demonstrate important contributions of both aerosol first and second indirect effects to the negative SCRE. Over Southern Mexico, although changes of CDNC and LWP are of comparable magnitudes between Gulf of Mexico and the land region (Fig.8), relative changes of both items are much larger over Gulf of Mexico (Fig. S6) due to the smaller magnitudes of background CDNC and LWP here (Fig. S5), which tends to lead to a more sensitive response of SCRE. That's why the maximum SCRE over Southern Mexico is more centered around Gulf of Mexico. Changes in ice water path (IWP) and ice crystal number concentration (ICNC) can also significantly affect SCRE, but with an opposite sign and mostly in the central U.S. The decreased IWP and ICNC, which are possibly caused by fire aerosol-induced changes in the circulation (Ten Hoeve et al, 2012 and reduced coarse mode dust aerosol concentrations), are responsible for the positive SCRE in the north part of central U.S. In the south part of central U.S., the reduction of IWP and ICNC also results in a positive SCRE, which partly offsets the negative SCRE resulting from changes in warm cloud properties. This explains the weaker total negative SCRE in this region compared to the Southern Mexico region despite the more substantial increase in CDNC and LWP here. "*

[Figure]

Figure S5. Spatial distributions of 10-day average (Apr. 1-10) ensemble mean a) column-integrated droplet number concentrations ($m^{-2}$) and b) liquid water path ($g\,m^{-2}$) in the E_NF simulations.

[Figure]

Figure S6. Relative changes of 10-day average ensemble mean cloud properties between the E_NF and E_QF simulations. a) cloud liquid water path, b) column-integrated droplet number concentration

**Minor comments:**

L60: studied->study

**Reply**: Done.

L370: Add "are" between "which" and "possibly";

**Reply**: Done.

L384: Same->The same;

**Reply**: Done.

L77-79: You should clarify why the indirect effect of fire aerosol deserves study.

Reply: Kaufman et al. (2005) and Zamora et al. (2006) show the short-term indirect effects of fire aerosols are strong based on satellite observations and aircraft measurements (Line 68-72 in the original manuscript). The fire aerosol indirect effect may significantly affect the cloud formation and radiative balance near wildfire burning region. We now explicitly mention the significant radiative effect of fire aerosol indirect effect in the previous paragraph (Line 65-66) to emphasize this as one motivation of our study. We further pointed out in this paragraph the current lack of model simulations of short-term fire aerosol indirect effects, which is another motivation of our work.

L101-103: What's the difference between nudging horizontal wind and temperature.

**Reply:** Nudging the horizontal winds will constrain the circulation towards reanalysis, but the thermodynamical features are not directly affected. If temperature is nudged strongly (i.e. use small relaxation time scale) too, the heating/cooling introduced by nudging may affect large scale vertical motion and the parameterized convection. In our study, horizontal wind nudging was applied to constrain the large scale circulations, thus a shorter relaxation time scale of 6 hour is adopted. On the other hand, we only used very weak temperature nudging (much longer relaxation time scale) and perturbed the nudging time scale gently to create ensembles. A much longer relaxation time scale of about 10 days is used. We have clarified this difference in the revised paper. Time scales of wind nudging and temperature nudging are now explicitly provided in the corresponding paragraph. The text reads (Line 92): "*Horizontal winds were nudged towards 6-hourly reanalysis to constrain the large-scale circulation*" and (Line 96): "*we also employed very weak temperature nudging (~10days) in combination with ensembles to quantify the uncertainty. More details of the nudging setup are described in section 2.3.*"

L297-503: How you can get the conclusion that at least 9 members are needed from Fig. 14. You need to quantify the results.

Reply: Thanks for the suggestions. The number 9 in the discussion paper was determined by simple visual comparison. As shown in Fig.14b, discrepancies in the ensemble mean fire aerosol SCRE are substantial when the number of ensemble members (N) is smaller than 8. We agree it is better to determine the minimum required ensemble number in a quantitative way. We now use results from the 20-member ensemble simulations as a reference to evaluate the results from ensemble simulations with varying N. For a specific N, the root mean square error (RMSE) of the ensemble mean SCRE during April 1-10 is chosen to quantify the deviation of the simulated ensemble mean from the reference value. It is calculated as the standard deviation of the differences between the daily ensemble mean SCRE in the N-member simulation and the 20- member simulation. To get robust results, for each N, we randomly sample N members from the 20 members for 1000 times and evaluated the performance of the 1000 groups (each group has N members) to avoid the influence of limited sampling. Figure 15 shows that both the RMSE of ensemble mean SCRE and the difference of RMSE between the 1000 groups of simulations (for each N) decrease with increasing N. The minimum number of N required is determined when the 90th percentile of RMSE is smaller than a threshold RMSE. Without a good reference, we set the threshold RMSE to 20% ($0.566\text{W m}^{-2}$) of the reference 10-day mean SCRE ($-2.83\text{Wm}^{-2}$). As shown in Fig.15, at least 11 members are needed to meet this criterion. We've refined the conclusion regarding the total number of ensembles needed in the revised paper. The corresponding paragraph now reads (Line 395-408):

*"However, discrepancies in the ensemble mean fire aerosol SCRE (Fig. 14b) are substantial when the number of ensemble members is small. The same is true for the ensemble spread of fire aerosol SCRE (Fig.S8).In order to quantify the discrepancies of the simulated SCRE, we chose the ensemble mean SCRE in the 20-member simulation as a reference and use the root mean square errors (RMSE) of the ensemble mean SCRE in the N-member simulation to quantify the deviation of the simulated SCRE from the reference value. It is calculated as the standard deviation of the differences between the daily ensemble mean SCRE in the N-member simulation and the 20-member simulation. For each N, we randomly sampled 1000 times from the 20 members to help reduce the influence from limited sampling. Figure 15 shows that both the RMSE of ensemble mean SCRE and the difference of RMSE between the 1000 groups of simulations (for each N) decrease with increasing N. The minimum number of N required is determined when the 90th percentile of RMSE is smaller than a threshold RMSE. Without a good reference, we set the threshold RMSE to 20% ($0.566\text{ W } m^{-2}$) of the reference 10-day mean SCRE ($-2.83\text{ W } m^{-2}$). As shown in Fig.15, at least 11 members are needed to meet this criterion."*

[Figure]

Figure 15 Root mean square errors (RMSE) of the ensemble mean of the regional mean fire aerosol SCRE during April 1-10 over Southern Mexico in simulations with different total number of ensemble members (N). The blue line represents the median RMSE of the 1000 groups (each group has N members/simulations). The grey line represents the threshold RMSE. Shaded area denotes the range between the 10th and 90th percentiles.

Reply to Anonymous Referee #2

We thank the reviewer for the helpful and constructive comments. Our point-by-point responses are provided below (reviewer's comments marked in blue and our responses in black.

This papers studies the direct/indirect aerosol effect from fires using CAM5 with nudged horizontal wind speed and/or nudged temperature. Overall the approach is sound and the paper is well written. Yet it still needs some major clarifications before it is accepted for publication.

**General comments:**

Since fire emission inventories are critical to this study, please provide a table or a plot to show the BC/OC/SO2 emissions from the 3 different inventories quantitatively. The colorbar in Fig. 2 is difficult to tell how bigger is the QF than the GF3/4. It seems QF emissions are at least 5 times larger than the other two. Please provide a table showing the different fire aerosol forcing components. In the introduction part, 3 different aerosol forcing are mentioned, but only direct forcing and short wave cloud forcing are presented in the result section. Please show long wave forcing and surface albedo forcing as well in the table. The initial condition for the 10-day ensemble runs is generated by S_NF with only u and v being nudged. Temperature is not nudged for the S_NF run. So my question is when you now include slow temperature nudging in the ensemble runs, will they go through some adjustment through the 10-day period? Or in other word, how well is the initial T compared to the T being nudged to on April 1st?

**Reply:** We have added the Table S1 to show the monthly mean and 10-day mean BC/OC/SO2 emissions from the three inventories. In both regions, the factor difference between the GFED and QFED data is similar no matter whether monthly mean or 10-day mean is used for comparison. Overall, the QFEDv2.4 emissions are larger than GFED v4.1s emissions by a factor of approximately 10 in the Central U.S and a factor of approximately 3 in Southern Mexico. We mentioned in the original manuscript that "*Values of emitted BC in QFED v2.4 are larger than those in GFED v4.1s by a factor of 9.7 in the Central U.S. and a factor of 2.7 in Southern Mexico*", but didn't specify this was based on the 10-day mean emissions. This has been clarified in the revised paper (Line 152-153).

Following the reviewer's suggestion, we have also added Table S3 to show the different components of fire aerosol effects. As shown in Table S3, Shortwave direct radiative effect (SDRE) and shortwave cloud radiative effect (SCRE) are the major contributors to the total fire aerosol effect in both regions. The surface albedo effect is almost negligible. The E_QF simulation shows a significant negative longwave cloud radiative effect (LCRE) in the Central U.S., which is associated with reduced ice crystal number concentrations and ice water path in the north part of this region. We have added a brief description of the negative LCRE (Line 344-345) in the revised paper. Causes of the fire aerosol effect are discussed in detail in Section 3.2.2.

The global pattern of the initial T (or the anomaly) is very similar to those in the reanalysis data but there are some regional differences. It should be noted that the horizontal wind nudging and temperature nudging are applied for different purposes. The horizontal wind nudging is used to constrain large-scale features in simulations with and w/o fire emissions to help better identify radiative forcing signal, while the weak temperature nudging conducted with gently perturbed time scale is used to generate ensemble and investigate the fire forcing uncertainty. Given its large relaxation time scale (10 to 11days compared to the 6-hourly wind nudging), the temperature nudging with small adjustments of temperature has little effect on constraining the temperature field towards reanalysis. We have added the time scale of wind and temperature nudging in Line 92 and Line 96 to clarify the difference.

Table S1 Regional mean emissions of fire aerosols in April, 2009 from three emission inventories (Unit: $x10^{-12}$ kg m$^{-2}$s$^{-1}$). Numbers in the parentheses show results averaged in April 1-10.

| | BC | | OC | | SO2 | |
|---|---|---|---|---|---|---|
| | Central U.S. | Southern Mexico | Central U.S. | Southern Mexico | Central U.S. | Southern Mexico |
| GFED v3.1 | 0.25(0.38) | 0.69(0.82) | 1.82(3.58) | 5.60(6.77) | 1.35(2.01) | 3.69(4.35) |
| GFED v4.1s | 0.23(0.34) | 1.17(1.44) | 1.75(3.24) | 8.80(10.76) | 1.21(1.81) | 6.25(7.69) |
| QFED v2.4 | 2.63(3.29) | 3.87(3.87) | 23.54(32.25) | 36.81(36.58) | 14.04(17.59) | 20.62(20.65) |

Table S3 Regional mean total AOD, fire AOD (differences in AOD between simulations with and without fire) and radiative effects of fire aerosols during April 1-10, 2009 in group B simulations (Unit: $W\,m^{-2}$). Total fire aerosol radiative effect is decomposed into shortwave direct radiative effect (SDRE), shortwave cloud radiative effect (SCRE), longwave cloud radiative effect (LCRE) and surface albedo effect (SAE).

| | Total AOD | Fire AOD | SDRE | SCRE | LCRE | Total SAE |
|---|---|---|---|---|---|---|
| Central U.S. | | | | | | |
| E_NF | 0.047 | | | | | |
| E_GF3 | 0.050 | 0.003 | 0.02 | -0.86 | 0.04 | 0.02 |
| E_GF4 | 0.050 | 0.003 | -0.01 | -0.39 | 0.002 | -0.003 |
| E_QF | 0.08 | 0.033 | -0.10 | -0.56 | -0.76 | 0.12 |
| Southern Mexico | | | | | | |
| E_NF | 0.135 | | | | | |
| E_GF3 | 0.149 | 0.014 | -0.18 | -1.91 | -0.21 | 0.06 |
| E_GF4 | 0.153 | 0.018 | -0.20 | -2.06 | -0.23 | 0.11 |
| E_QF | 0.202 | 0.067 | -0.86 | -3.02 | -0.47 | 0.14 |

**Specific comments:**

**Comment:** Abstract: why no forcing numbers are provided here. It is expected to see direct forcing and net indirect forcing rather than some changes in the short wave cloud forcing.

**Reply:** We have added forcing numbers in the revised paper and the text reads (Line 30-32 ) : "*Strong negative shortwave cloud radiative effect (SCRE) covers almost the entire Southern Mexico with a 10-day regional mean value of -3.02W $m^{-2}$. Over the Central U.S, comparable positive and negative SCRE of approximately 2W $m^{-2}$ appear in the north and south part of the region respectively.* "

**Comment**: Line 62-76: Please show what these forcings are. Direct or indirect?

**Reply:** We have specified direct or indirect effect (Lines 60-67).

**Comment:** Line 103: Please provide relaxation time for the very weak temperature nudging

**Reply:** The nudging time scale has been added and it now reads (Line 96): *"Even for short simulations, small perturbations of meteorological states might have large impact on the local aerosol and cloud properties, thus bring uncertainty to the aerosol forcing estimate. Therefore, in our simulations, we also employed very weak temperature nudging (~10days) in combination with ensembles to quantify the uncertainty. More details of the nudging setup are described in Section 2.3."*

**Comment**: Line 131-132: Please show or elaborate how you convert monthly mean emissions to daily emissions?

**Reply**: The GFED website (http://www.globalfiredata.org/data.html) provides monthly emission data and daily scalars to convert the monthly emission to daily emission. We have added following descriptions in Section 2.2 (Line 121-123): *"The daily emission data is obtained using daily scalars (http://www.globalfiredata.org/data.html) to distribute monthly emissions over the days and is only available from 2003 onwards."*

**Comment**: Line 138-142: Does the CAM5 default/background emission already include fire emission? Or did you remove fire emissions from the CAM5's emission files if there is any?

Reply: We didn't use the default climatological monthly mean fire emission in the CAM5 model. We replaced the original fire emission with the daily emission dataset mentioned in this study.

**Comment:** Line220-226: Please define these forcings with a few sentences rather than refer readers to Ghan 2013.

**Reply:** We have added a few sentences to elaborate the calculation of fire aerosol effect. Fire aerosol DRE, CRE and surface albedo effect are first defined as changes in total aerosol forcing, cloud forcing and surface albedo forcing between simulations with and without fire emissions. Then, a detailed description of the calculation of total aerosol forcing, cloud forcing and surface albedo forcing in the fire (or no fire) simulation is provided. The text now reads (Line 194-204)

*"Similar to Jiang et al. (2016), our calculations are based on the work of Ghan et al. (2012) and Ghan (2013). Fire aerosol DRE, CRE and surface albedo effect are defined as fire induced changes in aerosol forcing, cloud forcing, and surface albedo forcing respectively, and are calculated as the difference of each item between simulations with and without fire emissions (denoted by Δ). In each simulation, aerosol forcing was defined as the difference between all-sky and clean-sky TOA radiative fluxes ($F - F_{clean}$). Cloud forcing was defined as the difference between all-sky and clear sky TOA radiative fluxes under clean-sky conditions ($F_{clean} - F_{clean,clear}$). The rest were related to surface albedo forcing ($F_{clean,clear}$). Thus fire aerosol DRE, CRE, and surface albedo effect were expressed as $\Delta(F - F_{clean})$, $\Delta(F_{clean} - F_{clean,clear})$, and $\Delta F_{clean,clear}$, respectively. More details about the method can be found in section 2 of Ghan (2013). CRE includes contributions of both aerosol indirect effect and aerosol semi-direct effect but was analyzed as a single term (i.e., the sum)."*

**Comment**: Line 238: Please explain what "LEV 2.0 cloud-screened" is.

**Reply:** The AERONET web page (https://aeronet.gsfc.nasa.gov/) provides both original raw data and processed data at AERONET sites. The LEV 2.0 AOD is the processed data based on a cloud-screening algorithm (Smirnov et al. 2000). This information is now added (Line 214-215):
*"LEV 2.0 cloud-screened all points AOD at 500 nm and 675 nm was used to generate hourly AOD at 550 nm, which are the processed data based on a cloud-screening algorithm (Smirnov et al. 2000)"*

**Comment:** How much does the fire emitted aerosols contribute to the total AOD? It would be helpful to show some estimate of the contributions from fire emitted aerosols and other background aerosols. I realized you presented background AOD and fire AOD later in Fig. 6. But it would be more helpful if you can present some data here when you quote the need to increase the fire emissions by a factor of 1-3. And please explain why increasing the fire emission by a factor 1-3 could then make the simulated AOD large enough to compare with the reanalysis. It seems it is still unlikely to me.

We have added Table S2 showing total AOD, fire AOD and the contributions of fire AOD for reference in the revised paper. The following description is added (Line 245-246): "*Fire aerosols-*

*induced AOD increase accounts for 8.1% (S_GF3), 11.2% (S_GF4) and 48.8% (S_QF) of the background AOD (Table S2).*"

In the original manuscript, we quoted the need to increase the emissions by a factor of 1-3 mainly in order to show that the underestimation of AOD in simulations driven by GFED data is common. Thus the results in this study are consistent with previous studies. We didn't mean to imply that the scale factor from 1-3 can also be applied in this case. In fact, the scale factors are regionally-specific. In the work of Tosca et al. (2013), they derived the scale factors by computing the ordinary least squares regression between the simulated AOD and the observed AOD (MISR and MODIS) for those months over 1997-2009 that cumulatively contributed to 80% of regional fire emissions. In the selected four regions in their work including South America, South Africa, equatorial Asia and boreal North America, the scale factors range from 1-3. For the selected region and period in this study, larger scaler factors are needed to make simulated AOD in agreement with the NRL and MACC reanalysis data. However, it's worth noting that both reanalysis data shows a positive bias when compared to AERONET data especially in the Central U.S, while a good agreement is found between the simulated AOD in S_QF and AERONET observations (shown in Fig.4 in the original manuscript). Therefore a scale factor that brings the simulated AOD in the GFED forced simulations comparable to that in the QFED forced simulation might be more reasonable. We have performed sensitivity analysis by performing simulations driven by varying magnitudes of fire emissions (0, 1, 2, 5 times the emission). Results show that changes of fire AOD is proportional to changes in fire emissions. Therefore, the conclusion of a scale factor from 1-3 to match observations still holds in Southern Mexico, but a much larger scale factor of about 10 is needed in the Central U.S. We reorganized corresponding sentences in the revised paper to avoid ambiguity (Line 239-241): "*Previous studies have found the underestimation of AOD in simulations with GFED emissions and suggested the need to scale up GFED emissions by a factor of 1-3 to match the observed AOD (Tosca et al., 2013). This is consistent with the large negative bias in the simulation S_GF3 and S_GF4.*" and clarified that (Line 242): "*However, a much larger scale factor might be needed in this case.*"

Table S2 Regional mean total AOD, fire AOD (difference in total AOD between simulations with and without fire) and the contributions of fire AOD (fire AOD divided by total AOD in the S_NF simulation)during April, 2009 in group A simulations.

| | Central U.S. | | | Southern Mexico | | |
|---|---|---|---|---|---|---|
| | Total AOD | Fire AOD | Percentage | Total AOD | Fire AOD | Percentage |
| S_NF | 0.066 | | | 0.130 | | |
| S_GF3 | 0.068 | 0.002 | 3.42% | 0.141 | 0.011 | 8.10% |
| S_GF4 | 0.070 | 0.004 | 5.63% | 0.145 | 0.015 | 11.20% |
| S_QF | 0.099 | 0.033 | 49.33% | 0.194 | 0.064 | 48.84% |

**Comment**: Line 357-358: It is confusing here. Please consider revising.

**Reply**:  The text is revised as follows (Line 317-319): *"In the central U.S, despite moderate fire aerosol SDRE,  a positive fire aerosol SCRE exceeding 2 $W\ m^{-2}$ appears in the north part of the region while a comparable negative SCRE appears in the south part of the region."* Discussions on these results are provided in the next paragraph.

**Comment:** In the simulations with nudged U and V, the circulation is constrained. So it seems the circulation change may be small enough. Then use this to explain to change of ice clouds. I suspect the coarse mode dust number may be smaller and this may contribute to the decrease of produced ice number since the ice nuclei number (dust) is smaller. Need further investigation here.

Reply: We thank the reviewer for the helpful comment. Yes. The coarse mode dust concentration is reduced in the S_QF simulation (Figure S7), thus it is one of the possible causes for the reduced cloud ice amount and cloud ice number concentration. However, note that the location of maximum cloud ice decrease is a bit different from the location of maximum coarse mode dust concentration. In addition, a rough estimate of the change in coarse mode dust number concentration is on the order of $10^{-3}\ kg^{-1}$ when using dust aerosol density of about 2.5 $g\ cm^{-3}$ and a coarse mode diameter of about $2\mu m$. This means, even assuming that all the reduced coarse mode dust aerosols are effective ice nuclei, the resulting decrease in cloud ice number concentration from heterogeneous nucleation is still one order smaller than that shown in Figure S7 b).

Although horizontal wind is nudged in this study, changes in the vertical heating profile (e.g. due to BC absorption) would affect the distribution of large-scale vertical velocity. We notice a substantial change in the vertical velocity right below the location with maximum reduced ice cloud. Positive values in Figure S7e) represent downward motions induced by fire aerosols, which lead to a reduced vertical transport of moisture to the upper levels and will suppress the homogeneous ice nucleation there.

We have clarified this in the revised paper and the text now reads (Line 342-344): "*The decreased IWP and ICNC, which are possibly caused by fire aerosol-induced changes in the circulation (Ten Hoeve et al, 2012) and reduced coarse mode dust aerosol concentrations (Fig.S7), are responsible for the positive SCRE in the north part of central U.S*"

[Figure]

Figure S7. Pressure and longitude distribution of meridional mean (40-45 ° N) difference of 10-day average (April 1 -10) ensemble mean between simulation E_NF and E_QF: a) cloud ice amount $(kg \cdot kg^{-1})$ b) cloud ice number concentration $(kg^{-1})$ c) cloud fraction (1) d) Coarse mode dust concentration $(kg \cdot kg^{-1})$ e) vertical velocity $(Pa \cdot s^{-1})$ f) vertical moisture transport $(kg \cdot kg^{-1} \cdot Pa \cdot s^{-1})$

**Comment**: Line 385**:** make it clear it is SW cloud forcing

Reply: Corresponding sentences have been revised (Line 357-359).

**Comment**: Line 420: Why quote Fig. 3 here? I think Fig.3 shows results from Group A not from ensemble runs.

Reply: Fig.3 shows the results from both group A (for the whole month) and group B (for the first 10 days) simulations. Results from the ensemble runs are indicated by the range between the maximum and minimum values among ensemble members with darker colors. However, since the simulated AOD is barely distinguishable among members, and between two groups of simulations, the line and the shaded areas almost overlap. The shaded range can be more easily seen in Fig.3 b) by comparing the difference between first 10 days and the rest 20 days. We mentioned in the original manuscript that the shaded areas are very narrow. We have added following sentences in the caption of Fig.3 to further clarify this "*For the single-member simulation and the ensemble simulation driven by same fire emission, the shaded area and the solid line almost overlap given the barely indistinguishable AOD between ensemble members and the corresponding Group A simulation.*"

**Comment:** Line 430-432: How is the spread calculated for different N? Also how do you select the ensemble member for each different N? I suspect the number 9 required to converge may be different if the ensemble members for different N are constructed differently

Reply:

1) The spread is calculated as the sample standard deviation of ensemble members. For a specific N (number of ensemble members), values of ensemble members are denoted as Xi ( i = 1, 2, …N). Ensemble mean is calculated as $\overline{X} = \frac{1}{N}\sum_{i=1}^{N} Xi$. Ensemble spread is calculated as $\sigma = \sqrt{\frac{1}{N-1}\sum_{i=1}^{N}(Xi - \overline{X})^2}$ .

2) We performed a group of simulations with a total of 20 members. They are generated by implementing a weak temperature nudging with relaxation time scale ranging from (10 ~11.9 days with an interval of 0.1 day). For a specific N, the first N members are chosen in calculation, that is, members with relaxation time scale ranging from (10 ~ 10 + 0.1(N-1) days).

3) We thank the reviewer for this insightful comment, which has allowed us to further validate the robustness of our results. A quantitative way is now provided to determine the minimum numbers of ensemble members required in this case study. We now use results from the 20-member ensemble simulations as a reference to evaluate the results from ensemble simulations with varying N. For a specific N, the root mean square error (RMSE) of the ensemble mean SCRE during April 1-10 is used to quantify the deviation of the simulated ensemble mean from the reference value. It is calculated as the standard deviation of the differences between the daily ensemble mean SCRE in the N-member simulation and the 20-member simulation. To get robust results, for each N, we randomly sample N members from the 20 members for 1000 times and evaluated the performance of the 1000 groups (each group has N members) to avoid the influence of limited sampling. Figure 15 shows that both the RMSE of ensemble mean SCRE and the difference of RMSE between the 1000 groups of simulations (for each N) decrease with increasing N. The minimum number of N required is determined when the 90$^{th}$ percentile of RMSE is smaller than a threshold RMSE. Without a good reference, we set the threshold RMSE to 20% (0.566W m$^{-2}$) of the reference 10-day mean SCRE (-2.83 W m$^{-2}$). As shown in Fig.15, at least 11 members are needed to meet this criterion. We've refined the conclusion regarding the total number of ensembles needed in the revised paper. Corresponding paragraph has been rewritten and now reads (Line 395-408):

*"However, discrepancies in the ensemble mean fire aerosol SCRE (Fig. 14b) are substantial when the number of ensemble members is small. The same is true for the ensemble spread of fire aerosol SCRE (Fig.S8).In order to quantify the discrepancies of the simulated SCRE, we chose the ensemble mean SCRE in the 20-member simulation as a reference and use the root mean square errors (RMSE) of the ensemble mean SCRE in the N-member simulation to quantify the deviation of the simulated SCRE from the reference value. It is calculated as the standard deviation of the differences between the daily ensemble mean SCRE in the N-member simulation and the 20-member simulation. For each N, we randomly sampled 1000 times from the 20 members to help reduce the influence from limited sampling. Figure 15 shows that both the RMSE of ensemble*

*mean SCRE and the difference of RMSE between the 1000 groups of simulations (for each N)*

*decrease with increasing N. The minimum number of N required is determined when the 90th*

*percentile of RMSE is smaller than a threshold RMSE. Without a good reference, we set the*

*threshold RMSE to 20% (0.566$\mathrm{Wm^{-2}}$) of the reference 10-day mean SCRE (-2.83$\mathrm{W\ m^{-2}}$). As*

*shown in Fig.15, at least 11 members are needed to meet this criterion."*

[Figure]

Figure 15 Root mean square errors (RMSE) of the ensemble mean of the regional mean fire aerosol SCRE during April 1-10 over Southern Mexico in simulations with different total number of ensemble members (N). The blue line represents the median RMSE of the 1000 groups (each group has N members/simulations). The grey line represents the threshold RMSE. Shaded area denotes the range between the 10th and 90th percentiles.

Comment: Fig3: Please give full name of TCC.

Reply: Full name of TCC is temporal correlation coefficient. We now provide the full name in the revised paper.

Comment: Fig7: what is the KS test? Please give full name

Reply: Full name of the KS test is Kolmogorov-Smirnov test. We now provide the full name in the revised paper.

Comment: Fig10: is a) total aerosol direct forcing?

**Reply**: Yes. We have clarified this in the revised paper.

[revised manuscript text omitted]

---

## Author Response (AR2)

We thank the editor for the comments. Corrections have been made accordingly except for the following two.

For the comment "If I interpret Figure 15 correctly the graph indicates that only 9 ensembles are necessary (instead of 11 indicated in the paper) to satisfy the condition of RMS error < 0.58 W m-2"

As we mentioned in the manuscript, the minimum ensemble number is determined when the 90[th] percentile of RMSE is smaller than a threshold RMSE (indicated by the grey line). Here we use the 90[th] percentile of RMSE (indicated by the shaded area) instead of the median RMSE (indicated by the blue line). For this criterion, 11 is the minimum number required. Therefore, the original manuscript remains unchanged.

[Figure]

Figure 15 Root mean square errors (RMSE) of the ensemble mean of the regional mean fire aerosol SCRE during April 1-10 over Southern Mexico in simulations with different total number of ensemble members (N). The blue line represents the median RMSE of the 1000 groups (each group has N members/simulations). The grey line represents the threshold RMSE. Shaded area denotes the range between the 10[th] and 90[th] percentiles.

For the comment Change "For the selected region and period in this study, larger scaler factors are needed to make simulated AOD in agreement with the NRL and MACC reanalysis data." to "For the selected region and period in this study, larger scaling factors are needed to make simulated AOD in agreement with the NRL and MACC reanalysis data."

Change "Therefore, the conclusion of a scale factor from 1-3 to match observations still holds in Southern Mexico, but a much larger scale factor of about 10 is needed in the Central U.S."

to "Therefore, the conclusion of a scale factor from 1-3 to match observations still holds in Southern Mexico, but a much larger scaling factor of about 10 is needed in the Central U.S."

The two sentences appeared in author's "response to comment" instead of the manuscript. The wording error of "scaling factor" elsewhere in the manuscript has been corrected.